



# Dynamics of the Great Oxidation Event from a 3D photochemical-climate model

Adam Yassin Jaziri[1], Benjamin Charnay[2], Franck Selsis[1], Jérémy Leconte[1], and Franck Lefèvre[3]

[1]Laboratoire d'astrophysique de Bordeaux, Univ. Bordeaux, CNRS, B18N, allée Geoffroy Saint-Hilaire, 33615 Pessac, France
[2]LESIA, Observatoire de Paris, Université PSL, CNRS, Sorbonne Université, Université de Paris, 5 place Jules Janssen, 92195 Meudon, France.
[3]Laboratoire Atmosphères, Milieux, Observations Spatiales (LATMOS), CNRS/IPSL/UPMC, Paris, France.

**Correspondence:** Adam Yassin Jaziri (adam.jaziri@u-bordeaux.fr)

**Abstract.** From the Archean toward the Proterozoic, the Earth's atmosphere underwent a major shift from anoxic to oxic conditions, around 2.4 to 2.1 Gyr, known as the Great Oxidation Event (GOE). This rapid transition may be related to an atmospheric instability caused by the formation of the ozone layer. Previous works were all based on 1D photochemical models. Here, we revisit the GOE with a 3D photochemical-climate model to investigate the possible impact of the atmospheric circulation and

the coupling between the climate and the dynamics of the oxidation. We show that the diurnal, seasonal and transport variations do not bring significant changes compared to 1D models. Nevertheless, we highlight a temperature dependence for atmospheric photochemical losses. A cooling during the late Archean could then have favored the triggering of the oxygenation. In addition, we show that the Huronian glaciations, which took place during the GOE, could have introduced a fluctuation in the evolution of the oxygen level. Finally, we show that the oxygen overshoot which is expected to have occurred just after the GOE, was

likely accompanied by a methane overshoot. Such high methane concentrations could have had climatic consequences and could have played a role in the dynamics of the Huronian glaciations.

## 1 Introduction

The oldest rocks found today in the northwestern Canada date to 4.00-4.03 Gyr ago (Bowring and Williams, 1999). The stromatolites from the Barberton formation of South Africa and the Warrawoona formation of Australia dated to about 3.5

Gyr ago are accepted as a sign of life (Furnes et al., 2004; Awramik et al., 1983; Brasier et al., 2006). The microbial fossils dated to 2.6 Gyr ago found in the Campbell formation of Cape Province in South Africa are identifiable as cyanobacteria (Pierrehumbert, 2010) as many other evidences starting 2.8 Gyr ago (Nisbet et al., 2007; Crowe et al., 2013; Lyons et al., 2014; Planavsky et al., 2014; Satkoski et al., 2015; Schirrmeister et al., 2016). Cyanobacteria are known to produce oxygen by photosynthesis. Oxygenic photosynthesis was likely already developed before the Great Oxidation Event (GOE) that happened

around 2.4 to 2.1 Gyr ago. During this event, the amount of oxygen increased from less than $10^{-5}$ present atmospheric level (PAL) to a maximum of $10^{-1}$ PAL around 2.2 Gyr ago before stabilising approximately at $10^{-2}$ PAL (Lyons et al., 2014).





The best constraints on the GOE come from sulfur isotopic ratios in precambrian rocks (Farquhar et al., 2007; Lyons et al., 2014). In the Archean anoxic atmosphere, the sulfur photochemistry was responsible for mass-independent fractionation of sulphur isotopes in sedimentary rocks (Kasting et al., 1989). The loss of mass-independent fractionation in sedimentary rocks

less than 2.5 Gyr ago is explained by the disappearance of sulfur photochemistry due to the rise of the amount of oxygen and to the formation of the ozone layer (Zahnle et al., 2006). $\sim$0.01 PAL of oxygen is sufficient to generate enough UV shielding by ozone to block the sulfur photolysis (Zahnle et al., 2006).

The GOE represents a major event in the history of the Earth. It profoundly impacted the atmospheric and oceanic chemistry, the climate, the mineralogy and the evolution of life. $O_2$ was a poison for a lot of anoxygenic form of life supposed already

developed. Consequently, the GOE likely induced a mass extinction for anoxygenic form of life, including heterotrophic methanogens (i.e. organisms producing methane). Methane may have been more abundant in the anoxic Archean atmosphere than today (1.8 ppm), with levels as high as $10^4$ pmm according to Catling and Zahnle (2020). Such high methane concentrations would have produced a significant greenhouse effect. The decrease of the biological methane productivity and the methane photochemical lifetime could have reduced its abundance and thus its warming contribution, potentially triggering

the Huronian glaciations that took place between 2.4 and 2.1 Gyr. The GOE is a key event to understand the co-evolution of life and environment on Earth but also on exoplanets. However, major questions remain concerning the triggering and the dynamics of the GOE.

Before the appearance of oxygenic photosynthesis, the redox state of the atmosphere was ruled by the balance between reductant fluxes from volcanism and metamorphism and the hydrogen atmospheric escape (Catling et al., 2001). The appearance

of oxygenic photosynthesis, which was much more efficient than previous metabolisms, relying on ubiquitous $H_2O$, $CO_2$ and light, profoundly changed the biogeochemical cycles. The oxygen is produced by oxygenic photosynthesis (summarized by formula (1)), which can be reversed by aerobic respiration.

$$CO_2 + H_2O + h\nu \longrightarrow CH_2O + O_2 \tag{1}$$

The burial of organic carbon (a very small fraction of the net primary productivity) allows the accumulation of oxygen until that

an equilibrium is reached between the burial of organic carbon, the reductant fluxes, the oxidative weathering (i.e. the oxidation of buried organic carbon re-exposed to the surface by plate tectonics) and the hydrogen escape. Assuming a methane-rich atmosphere, atmospheric oxygen is also strongly coupled to methane through the reaction of methane oxidation:

$$CH_4 + 2O_2 \longrightarrow CO_2 + 2H_2O \tag{2}$$

On the early Earth and once oxygenic photosynthesis appeared, methane would have been mostly produced by the fermentation

of organic matter followed by acetogenic methanogenesis:

$$2CH_2O \longrightarrow CH_3COOH \longrightarrow CH_4 + CO_2 \tag{3}$$

For aerobic conditions, it could have been consumed by oxygenic methanotrophy, similar to reaction (2). Goldblatt et al. (2006) and Claire et al. (2006) developed simplified models of the biogeochemical cycles of $O_2$ and $CH_4$. They proposed scenarios for the evolution of the amount of oxygen and methane and the dynamics of the GOE.



Several hypotheses have been proposed to explain the mechanisms and timing of the GOE. The hydrogen escape is one of them, proposed by Catling et al. (2001). Considering the irreversible oxidation of methane by the reaction (4) :

$$CH_4 + O_2 \longrightarrow CO_2 + 4H(\uparrow) \tag{4}$$

and the reverse reaction (2), we get a chain of reaction that causes a net gain of oxygen by transforming $2H_2O$ into $O_2$ (5) :

$$CO_2 + 2H_2O \longrightarrow CH_4 + 2O_2 \longrightarrow CO_2 + O_2 + 4H(\uparrow) \tag{5}$$

Emergence of continents and subaeriel volcanism is another hypothesis developed in Gaillard et al. (2011) that led to a change of the chemical composition and the oxidation state of sulfur volcanic gases, precipitating the atmospheric oxygenation.

But whatever precipitated the GOE, the rise of oxygen seems to be linked to an atmospheric instability caused by the formation of the ozone layer and its impact on the photochemical methane oxidation. Slowly increasing $O_2$, by oxygenic photosynthesis, would have accumulate enough starting the ozone layer formation. The ozone layer provide a photochemical
shield which limit the oxygen photochemical destruction leading to the methane oxidation. Therefor the oxygen could have accumulate more easily, producing more ozone, shielding more efficiently the oxygen destruction and then rising an instability of growing oxygen until others processes would have limited the oxygen abundance, such as rock oxidation.

In this paper we focus on the atmospheric photochemical losses by methane oxidation associated to the GOE. Previous 1D
studies of Goldblatt et al. (2006); Claire et al. (2006); Zahnle et al. (2006), developed dynamical models of the GOE based on 1D photochemical models. In this study, we use for the first time a 3D Global Climate Model to compute the chemical lifetime of the different species and to explore 3D effects. Since the oxygen build-up is linked to the formation of the ozone, we could expect effects from the latitudinal/longitudinal ozone distribution or from the variations of UV irradiation by the seasonal and the diurnal cycles. We also investigate the potential links between the GOE and the Huronian glaciations. In particular, the
consequences of a cold climate (.e.g. a snowball Earth event) on the photochemistry have never been studied.

Following, we describe the atmospheric model used for this study in Section 2. In Section 3, we analyze the photochemistry of the late Archean/Neoproterozoic with the 3D model, highlighting the chemical lifetimes and the impact of the global mean surface temperature. Based on these results, we describe the dynamical evolution of $O_2$ and $CH_4$ during the GOE in Section 4,
highlighting consequences of the Huronian glaciations. We finish with a summary and perspectives in Sect. 5.

## 2 Model

A 3D photochemical global climate model is used to characterize photochemical oxygen losses in the atmosphere, dominated by methane oxidation in the model. The model, the LMD-generic, is a generic Global Climate Model (GCM) initially developed at the Laboratoire de Météorologie Dynamique (LMD) for the study of a wide range of atmospheres. It allows to model easily
different atmospheres, which makes it widely used, for instance to study early climates in the solar system (Forget et al., 2012;





D. Wordsworth et al., 2012; Charnay et al., 2013; Turbet et al., 2020b) or extrasolar planets (Selsis et al., 2011; Leconte et al., 2013; Bolmont et al., 2016; Fauchez et al., 2019; Turbet et al., 2020a). From the photochemical module of the martian version of the model (Lefèvre et al., 2004), we have develop a generic version. It allows to easily adapt the chemical network and it introduces the calculation of the photolysis rates and their heating rates within the model.

The chemical network is derived from the REPROBUS model of the present Earth stratosphere (Lefèvre et al., 1998) but adapted to the assumed composition. Halogen, heterogeneous and nitrogen chemistry is not taken into account due to the weak constraints available and considering its negligible effect on the oxygen chemistry studied. In contrast, the chemical network includes a detailed methane chemistry. It allows to take into account the different pathways of the methane oxidation balance. The methane network is built according to Arney et al. (2016) and Pavlov et al. (2001). The whole chemical network is detailed

in appendix A.

We have developed a new photochemical module for the LMD-generic code. Although previous versions of the code already included photochemical processes, they were hard-coded to specific atmospheres. The module is now flexible and no longer uses pre-computed photolysis rates, which are now computed using the actual absorbers abundances in the model. The module also accounts for the heating rates by photolyses although the abundance of $O_3$ in the present study is too low to yield significant

heating. Including the heating by photodissociations will nevertheless be essential for other potential applications of the generic model, including Earth-like oxygen-rich atmospheres.

The GCM is adapted to the supposed conditions of the Archean Earth. The rotation period is set to 17.5 hrs (adapted for an orbital period of 500 days according to Zahnle and Walker (1987); Bartlett and Stevenson (2016)). The spectrum of the star is calculated for 2.7 Ga and 1.0 AU from Claire et al. (2006) (see Figure 1). We define an atmosphere with 98% of $N_2$ and 1% of

$CO_2$ for 1 bar at the surface. The topography of the Archean Earth presumes a central continent. We define then a ocean planet with an equatorial supercontinent as in Charnay et al. (2013) (latitude $\pm$ 38°, longitude $\pm$ 56°).

The 1D version of the model uses a surface ocean, a surface mean albedo of 0.28, a mean solar zenith angle of 60° and an eddy diffusion coefficient from Zahnle et al. (2006).

Photochemical losses from the atmosphere by methane oxidation in the GCM are balanced with a production flux at the

surface. This flux is established by fixing the abundance of the species considered in the first layer of the GCM (formula (8)). When the stationary regime is reached, we can quantify the total photochemical loss/production of a species as its integrated surface flux required to maintain constant its surface abundance. Several simulations are performed on a grid of oxygen and methane abundance at the surface. The calculated fluxes are used to determine the photochemical loss flux as a function of the variable oxygen and methane abundances at the surface.

The GCM ensures the convergence of carbonaceous species by also fixing the $CO_2$ abundance at the surface to 1%. In addition, the GCM takes into account atmospheric escape according to the Catling et al. (2001) model. The $H_2$ abundance in the last layer of the GCM is updated according to the formula 6 thanks to the escape flux calculated by the formula 7.

$$H_2^{top} = H_2^{top} - \frac{\Delta t \times F_{H_2}}{\Delta^{n-1->n}z \times dens^{top}} \tag{6}$$



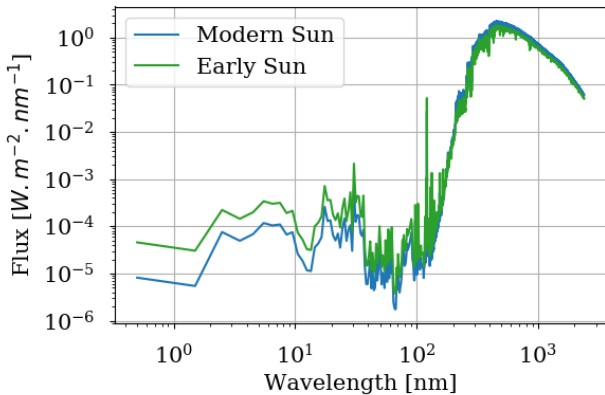

**Figure 1.** Stellar flux received by modern Earth computed for 0.0 Ga and 1.0 AU and stellar flux received by early Earth computed for 2.7 Ga and 1.0 AU from Claire et al. (2012).

$$F_{H_2} = 2.5 \times 10^{17} \times (H_2^{top} + 2CH_4^{top} + H_2O^{top}) \tag{7}$$

$$F_{O_2} = (O_2^{const} - O_2^{surf})\frac{\Delta^{1->2}z}{\Delta t}dens^{surf} \tag{8}$$

| | |
|---|---|
| Flux (surface or escape) | $F_i$ [molecules.m$^{-2}$.s$^{-1}$] |
| Given surface value | $O_2^{const}$ [ppmv] |
| First layer value | $O_2^{surf}$ [ppmv] |
| Last layer value | $H_2^{top}$ [ppmv] |
| Thickness of first layer | $\Delta^{1->2}z$ [m] |
| Thickness of last layer | $\Delta^{n-1->n}z$ [m] |
| Physical timestep | $\Delta t$ [s] |
| Surface density | $dens^{surf}$ [molecules.m$^{-3}$] |

## 3   Methane oxidation fluxes

The atmospheric oxygen loss is dominated by the oxidation of methane, formula (2). The oxidation of methane is catalyzed by OH radicals (Gebauer et al., 2017). These radicals are produced by the photochemistry of water vapor:

$$H_2O + h\nu \longrightarrow OH + H \tag{9}$$





$$H_2O + O(^1D) \longrightarrow OH + OH \qquad (10)$$

The amount of water vapor in the troposphere is controlled by temperature in a 1D model with an infinite water reservoir on the surface but in a 3D dynamic model with dry continental surfaces, it also depends on the horizontal transport, evaporation and precipitations.

Photochemical processes depend on insolation and therefore on diurnal and seasonal variations. The formation of the ozone
layer is a turning point for the photochemical balance. The ozone layer produces an UV shield, which limits the photochemical processes leading to methane oxidation and destroying oxygen. Oxygen can accumulate more efficiently and form more ozone. This positive feedback creates an oxygen instability which accounts for the sudden oxygenation of the atmosphere and may therefore be sensitive to the spatial distribution of ozone, and thus to the global 3D transport.

In this section, we compare the results between the 1D and 3D model on photochemical oxygen and methane losses at
135 steady state. We compute the variation of vertical chemical pathways to methane oxidation as a function of the surface $O_2$ fixed abundance. We also analyze the spatial distribution of ozone. Finally, we examine how surface fluxes required to sustain a steady state depend on surface temperature.

### 3.1 From 1D to 3D models

We ran the 1D model until steady state for a range of $O_2$ vmr from $10^{-7}$ to $10^{-3}$ and $CH_4$ vmr from $10^{-6}$ to $10^{-3}$. Figure 2
shows the total atmospheric $O_2$ loss ($F_{O_2}$) as a function of these two parameters. These results are consistent with the previous study of Zahnle et al. (2006).

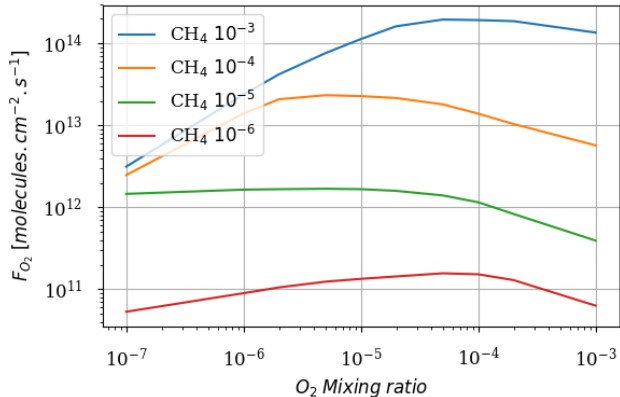

**Figure 2.** Oxygen atmospheric loss ($F_{O_2}$) as a function of surface $O_2$ for a surface $CH_4$ from $10^{-6}$ to $10^{-3}$.

Figure 3 shows the atmospheric losses of oxygen ($F_{O_2}$) and methane ($F_{CH_4}$), the hydrogen escape flux ($F_{H_2}$) and the ozone column density computed with the 1D and 3D version of the GCM, for a methane abundance of $10^{-4}$ and an oxygen abundance





grid between $10^{-7}$ and $10^{-3}$. The results are obtained after the convergence of the model with less than 1% variation of the

average last 50 time steps in 1D, last year in 3D compared to the previous one. This is done for the $H_2$, $CO_2$, $CH_4$ and $O_2$ surface flux, the $H_2$, $CO_2$, $CH_4$, $O_2$ and $O_3$ column density, the surface temperature and the Outgoing Longwave Radiation (OLR) and the Absorbed Shortwave Radiation (ASR). The surface fluxes at steady state in 3D present horizontal and seasonal variations and are averaged over the planetary surface and over one year (500 days). Timescales of the rise of oxygen are much larger than a year and the seasonal fluctuations are therefore not included in the following discussions. Discrepancies between

3D and 1D never exceed 10% for $F_{O_2}$ and $F_{CH_4}$. However, the ozone column is always found larger in 1D, up to 5 times the mean column obtained with 3D. The average profiles of $O_2$, $O_3$, CO, $CH_4$, $H_2$ and water vapor found in 1D and 3D are presented in Figure 6. Differences can come either from averaging the UV irradiation geometry in 1D or from horizontal and vertical transport. A priori, the photochemical losses ($F_{O_2}$ and $F_{CH_4}$) are not significantly modified (Figure 3) and the vertical transport seems responsible for these differences. The 3D vertical transport seems to transport species more efficiently than

the 1D transport model which uses an Eddy coefficient from Zahnle et al. (2006) to mimic the 3D transport. In particular, this results in smaller vertical gradients with the 3D model (Figure 6).

     Despite of the aforementioned small departures, we find that the 1D model reproduces the results of the more comprehensive 3D model.

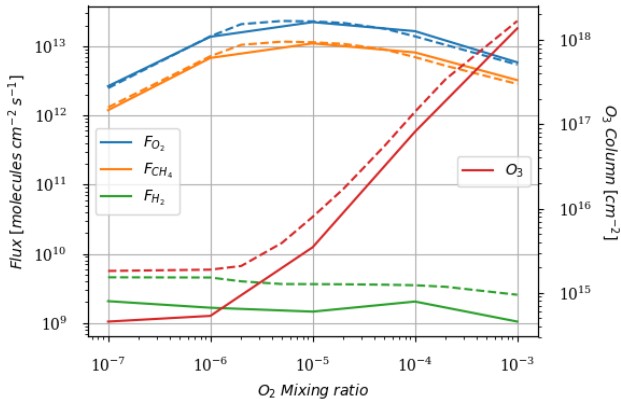

**Figure 3.** Oxygen atmospheric loss ($F_{O_2}$), methane atmospheric loss ($F_{CH_4}$), hydrogen escape ($F_{H_2}$) and $O_3$ column density as a function of surface $O_2$ for a surface $CH_4$ set up at $10^{-4}$. Solid: results of 3D model (averaged over the surface and a year). Dashed: 1D results.

### 3.2    Vertical distribution of $O_2$ losses

At steady state (averaging seasonal variations) $O_2$ photochemical losses compensate for the surface outgassing of $O_2$. The ratio 1/2 between $F_{O_2}$ and $F_{CH_4}$ (Figure 3) shows that the atmospheric losses of oxygen and methane are dominated by the methane oxidation reaction 2, which uses two molecules of $O_2$ for each molecule of methane (and forms one molecule of $CO_2$ and 2 molecules of $H_2O$). While oxygen molecules are mainly involved with CO and $CO_2$ cycle (Gebauer et al., 2017), $CH_4$





is dominated by the methane oxidation. So, to analyze how losses of molecular oxygen are distributed in altitude, it is simpler
to look at the methane loss, dominated by the oxidation of methane.

The Figure 4 represents the rate of methane destruction as a function of altitude and for different $O_2$ levels. As for their
integrated value, the vertical profiles of $F_{O_2}$ and $F_{CH_4}$ are similar when computed with 1D and 3D models are similar. We
distinguish the contribution to the loss of three main altitude domains : the troposphere, stratosphere and above. The losses
are dominated, whatever the $O_2$ abundance, by the tropospheric contribution. In the appendix B, we identify the main reaction
pathways leading to a net methane oxidation 2. Figure 4 shows a migration of losses from the troposphere to the stratosphere
when oxygen abundance increases. Catalysis by OH remains at the heart of the oxidation mechanism although a different
reaction pathway is identified (Appendix B): as the stratospheric ozone becomes more abundant, the production of $O(^1D)$
through its photolysis increases:

$$O_3 + h\nu \longrightarrow O_2 + O(^1D) \qquad\qquad (11)$$

which then initiates the production of OH through the reaction 10 instead of the $O(^1D)$ coming from $O_2$ photolysis. The
stratospheric contribution is less efficient than the tropospheric one, that is why $F_{O_2}$ decreases for the high abundances of
oxygen. Finally, there are $CH_4$ and $O_2$ losses in the upper atmosphere (around 10 Pa), which are less sensitive to the surface $O_2$
level. This contribution is no longer dominated by the catalysis of OH but comes from the photolysis of methane. The different
pathways are identified in appendix B.

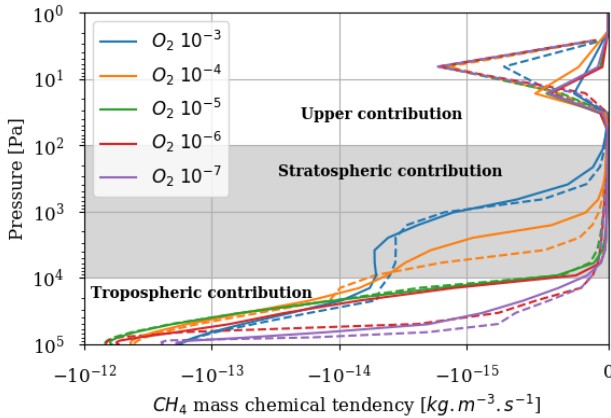

**Figure 4.** Methane photochemical losses as a function of altitude for several $O_2$ levels and a methane vmr of $10^{-4}$ at the surface. Solid:
results of 3D model (averaged over the surface and a year). Dashed: 1D results.

### 3.3 Ozone

The 3D effects have no significant consequence on the photochemical losses of $O_2$. The decrease in $F_{O_2}$ with increasing
levels of $O_2$ above an $O_2$ vmr of $10^{-5}$ (Figure 3) is due to the formation of the ozone layer, due to UV shielding of the





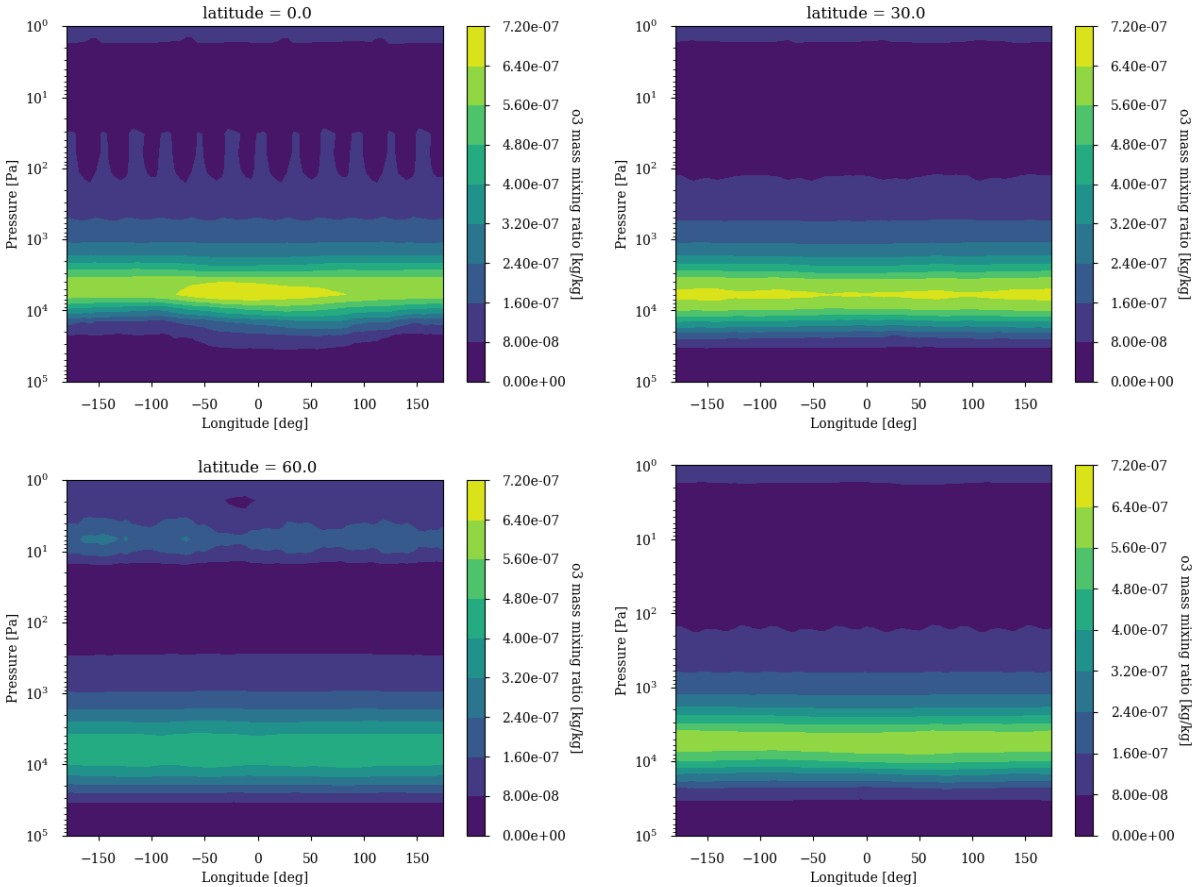

**Figure 5.** Ozone annual average longitudinal profile for latitude $0°$, $30°$, $60°$ and latitudinal average.
Surface: $CH_4 = 10^{-4}$ and $O_2 = 10^{-3}$

methane oxidation (Goldblatt et al., 2006; Zahnle et al., 2006). Figure 5 shows that ozone is relatively homogeneous over the whole planet making 1D modeling relevant. Nevertheless, there are some variations of the $O_3$ column with latitude and related variations of the biologically harmful UV flux reaching the surface (Figure 7). This non-homogeneous opacity of the ozone UV shield may be important for the evolution and distribution of organisms living at the surface.

### 3.4 Surface temperature effect

The tropospheric temperature profile follows a moist/dry adiabat that controls the vertical profile of water vapour, which, in turn, affects the greenhouse warming. Since this water vapor is the source of OH, catalyzer of methane oxidation, this interplay suggests a possible link between surface temperature and photochemical losses of $CH_4$ and $O_2$, which we investigate here.

The previous results (Figure 3) were obtained with a surface temperature close to 280 K, whatever the oxygen abundance at the surface and an abundance of methane of $10^{-4}$. The surface temperature is the same because all the parameters are the same





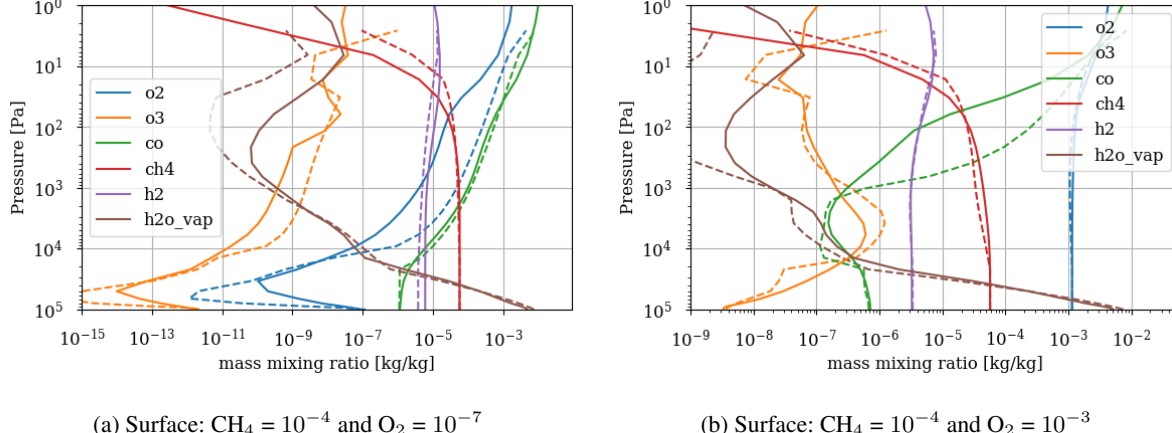

(a) Surface: $CH_4 = 10^{-4}$ and $O_2 = 10^{-7}$        (b) Surface: $CH_4 = 10^{-4}$ and $O_2 = 10^{-3}$

**Figure 6.** Species profile for 3D surface and annual average (solid) and 1D (dash) model.

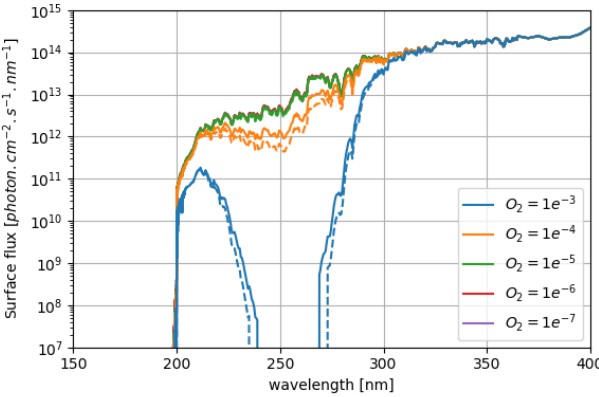

**Figure 7.** Stellar flux reaching the surface for several surface oxygen and surface methane of $10^{-4}$. Results of 3D model surface and annual average (solid) and 1D model (dash).

(rotation period, obliquity, solar input, surface composition (water/continent), continental albedo, etc...) and main greenhouse gases ($CO_2$, $CH_4$) are not changed depending on the oxygen level.

As a first test to assess the effect of surface temperature, we use the 1D model with a surface surface temperature forced to remain at 220 K (by setting to zero the heating/cooling rate of the surface). This is the approximate surface temperature that would be reached by the 1D model with a frozen start, although the actual value would depend on the level of greenhouse $CH_4$. This way we can evaluate the impact of a snowball event on the photochemistry. Figure 8 shows that photochemical losses decrease with the temperature. How strong this decrease is depends on the $O_2$ level, the drop being the largest around the

maximum of $F_{O_2}$, so for a vmr of $O_2$ around $10^{-5}$. At this $O_2$ level, the photochemical losses occur entirely in the troposphere. Figure 9 compares the vertical profiles of methane photochemical losses for a surface temperature of 280 K and 220 K and





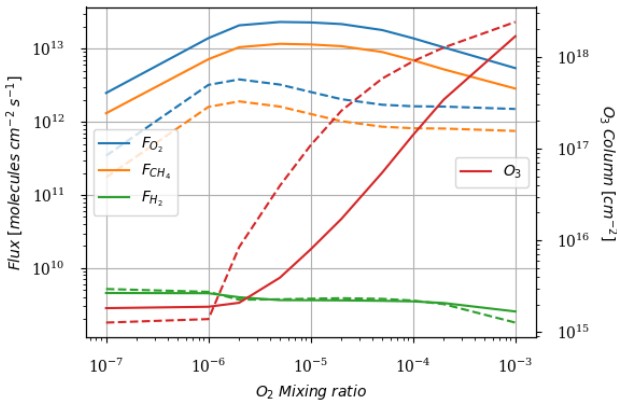
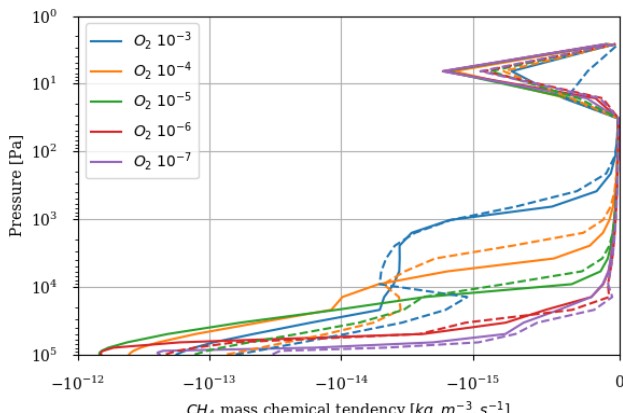

**Figure 8.** Surface oxygen ($F_{O_2}$) and methane ($F_{CH_4}$) fluxes, hydrogen escape ($F_{H_2}$) and $O_3$ column density as a function of surface $O_2$ for a surface $CH_4$ set up at $10^{-4}$. Results for 1D model surface temperature 280K (solid) and 220K (dashed).

**Figure 9.** Methane photochemical losses as a function of altitude for several $O_2$ levels and a surface $CH_4$ vmr of $10^{-4}$. Results from 1D modeling and for a surface temperature of 280K (solid) and 220K (dashed).

for different oxygen abundances at the surface. We see that the influence of surface temperature on the losses is located in the troposphere. Thus, the larger the tropospheric contribution, the larger the decrease in losses. The stratospheric thermal profile and the troposause (the cold trap that controls the transport of water vapor to the stratosphere) show little response to a decrease

of surface temperature from 280 K to 220 K (see Figure 12). As a consequence only the tropospheric chemistry is affected.

The main implication of this result is that, under conditions of a global glaciation, the oxygen instability is triggered for an oxygen abundance/flux about an order of magnitude lower. For a given $O_2$ flux, glacial conditions should favor the switch from an oxygen-poor to an oxygen-rich atmosphere. During glaciations, however, environments able to provide both light and liquid water are considerably limited making likely a considerable drop in the photosynthetic production of $O_2$ and the burial

of biomass.

The temperature trend of oxygen losses is determined. These losses are calculated using the 1D model by setting different surface temperatures for a methane abundance of $10^{-4}$ and an oxygen abundance of $10^{-5}$ at the surface. Figure 11 shows $F_{CH_4}$ (equivalent to $\frac{1}{2}F_{O_2}$) as a function of surface temperature. We observe a growth of losses of the order of one decade per 50 K. The discontinuity at 273 K is artificially produced by the change of albedo at the surface between ice-free oceans

(albedo = 0.07) and fully ice-covered oceans (albedo = 0.65). The increase in albedo below 273 K reflects more UV flux and increases photochemical losses in the troposphere. The 3D model smooths this effect by gradually freezing the ocean at the surface. We use the slow convergence of the 3D model towards a frozen state. The photochemical equilibrium is established on a time scale of a few years, whereas the progressive freezing of the surface takes place over several decades. A quasi-stationary state of species abundance in the atmosphere and consequently of atmospheric losses is rapidly established. During freezing

on a longer time scale, the evolution of temperature and oxygen losses in their quasi-stationary state is recorded to establish



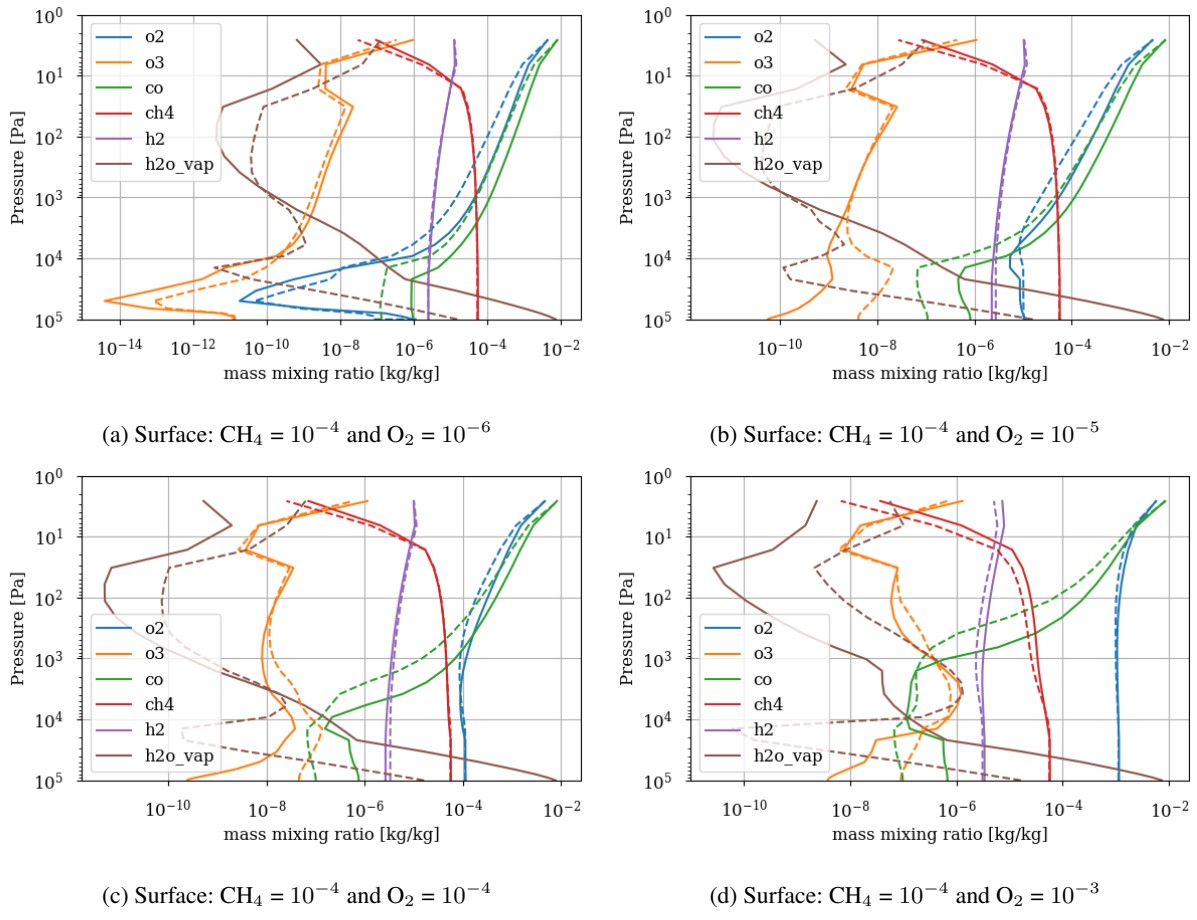

(a) Surface: $CH_4 = 10^{-4}$ and $O_2 = 10^{-6}$

(b) Surface: $CH_4 = 10^{-4}$ and $O_2 = 10^{-5}$

(c) Surface: $CH_4 = 10^{-4}$ and $O_2 = 10^{-4}$

(d) Surface: $CH_4 = 10^{-4}$ and $O_2 = 10^{-3}$

**Figure 10.** Species profile for 1D model surface temperature 280K (solid) and 220K (dashed).

the link between oxygen losses and temperature, see Figure 11. The 3D model converges to a frozen state where the sea ice extends from the poles to 20-25°N/S. The coverage is about 60-65%, as observed in Charnay et al. (2013), and the surface temperature converges to 230 K. A cold start with a completely frozen surface is then performed to evaluate the impact of the 3D model following a global glaciation. The results Figure 11 show that in addition to the smoothing of the albedo effect, the

trend is weaker than the 1D model. At the root of this difference is the difference in transport between the two models. The 3D model mixes the tropospheric temperature more efficiently, warming the troposphere and reducing the impact of a cooling on atmospheric losses, see Figure 12.

Temperature appears to be a variable in atmospheric losses that has a significant impact. Between a temperate state and an ice age, oxygen losses can vary by a factor of 2 to 4. Oxygen abundance and methane abundance can accumulate more in the

atmosphere during an ice age. The consequences of ice ages on the temporal evolution of oxygen and methane during the GOE are studied in the following by considering this new trend.





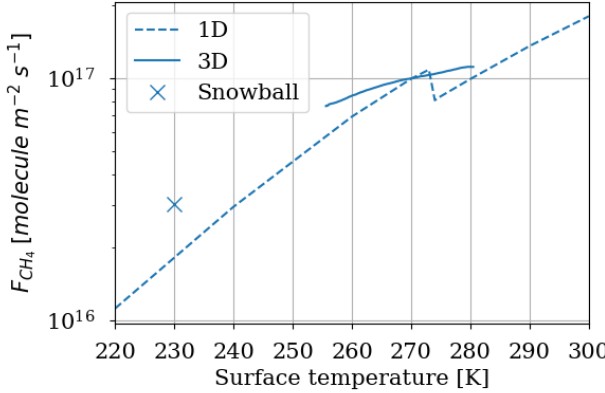

**Figure 11.** $F_{CH_4}$ depending on the surface temperature. Results of 3D model surface and annual average (solid and mark for snowball) and 1D model (dash). Surface: $CH_4 = 10^{-4}$ and $O_2 = 10^{-5}$.

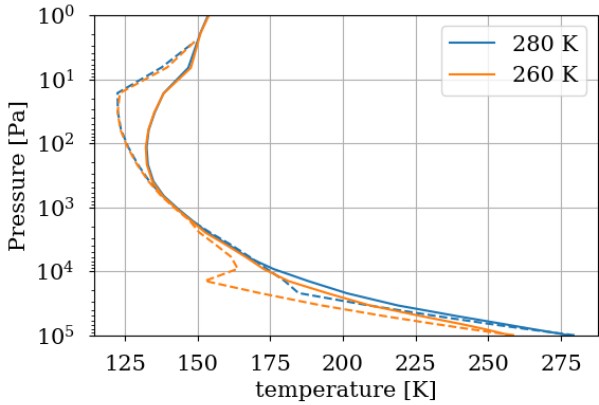

**Figure 12.** Temperature profile for a surface temperature of 280 K and 260 K. Results of 3D model surface and annual average (solid) and 1D model (dash).

## 4  Temporal evolution, overshoot and glaciation events

After the GOE, a carbon-13 isotopic variation of nearly 15 ‰ is observed (Lyons et al., 2014). This event is called the Loma-gundi event (Bachan and Kump, 2015). Although the dynamics of the oxygenation process remain uncertain, this event suggests
an over-oxygenation of the atmosphere (Catling and Zahnle, 2020). Other evidence supports this phenomenon, such as the evolution of the $\delta^{34}S$ fraction of carbonate-associated sulfates (Planavsky et al., 2012; Schröder et al., 2008) or fluctuations in the degree of uranium enrichment in organic-rich shales (Partin et al., 2013). An increase in overall oxygen productivity followed by a decrease would seem to account for this over-oxygenation (Harada et al., 2015; Holland and Bekker, 2012; Hodgskiss et al., 2019). Nevertheless, we have seen previously that a global glaciation phenomenon can significantly decrease the atmo-





spheric oxygen losses. It is therefore not impossible that Huronian glaciations could also have had an impact on atmospheric
over-oxygenation during the global oxygenation process. In order to study the dynamics of over-oxygenation, a time evolution
equation model is established based on the equations of Goldblatt et al. (2006), Claire et al. (2006) and adjusted to take into
account variations in primary oxygen productivity and atmospheric losses. We compare the previous Goldblatt et al. (2006)
parametrisation for atmospheric loss with the GCM interpolated function on the time evolution without an over-oxygenation

using the time evolution equation model established. We then apply this model to reproduce the over-oxygenation and the
fluctuations brought by glaciations.

## 4.1 Equations model

The temporal evolution of oxygen and methane abundance during the GOE is modeled by the Goldblatt et al. (2006) equations.
They relate atmospheric losses, described in section 3, to surface contributions associated with biogeochemical exchanges.

Goldblatt et al. (2006) propose for this biosphere model a parameterization of atmospheric losses according to $\Psi_{O_2}[CH_4]^{0.7}$
where $\Psi_{O_2} = 10^{a_1\psi^4+a_2\psi^3+a_3\psi^2+a_4\psi+a_5}$, $\psi = log([O_2])$, $a_1 = 0.0030$, $a_2 = -0.1655$, $a_3 = 3.2305$, $a_4 = -25.8343$ and $a_5 =$
71.5398. Goldblatt et al. (2006) also estimate the value of the different parameters (see Table 1). A set of values is associated
with a steady state of oxygen and methane abundances. Depending on the value of the flux of reductant $r$, one is on a state of
oxygen-poor or oxygen-rich equilibrium. In order to study the temporal evolution between these two equilibrium states, we

use the results of Claire et al. (2006) to establish a temporal evolution of the flux of reductant $r$ (Figure 13). We introduce this
temporal evolution in the equations of Goldblatt et al. (2006) to reproduce the dynamics of oxygenation. Finally, we introduce
a coefficient $\alpha_N$ ($\geq 1$) to model a photosynthetic over-productivity responsible for the over-oxygenation, as well as a coeffi-
cient $\alpha_\Psi$ ($\leq 1$) to model the decrease of the atmospheric losses during a glaciation. The evolution of the abundance of oxygen,
methane and buried carbon is then described by equations:

$$260 \quad \frac{d[CH_4]}{dt} = \alpha_N \frac{1}{2}\Omega_{O_2}N + \frac{1}{2}\Omega_{O_2}r - s[CH_4] - \alpha_\Psi \frac{1}{2}\Psi_{O_2}[CH_4]^{0,7} - \frac{1}{2}\Omega_{O_2}(\beta(N+r) - wC) \tag{12}$$

$$\frac{d[O_2]}{dt} = \alpha_N\Omega_{O_2}N - (1-\Omega_{O_2})r - s[CH_4] - \alpha_\Psi\Psi_{O_2}[CH_4]^{0,7} - (1-\Omega_{O_2})(\beta(N+r) - wC) \tag{13}$$

$$\frac{dC}{dt} = \beta(N+r) - wC \tag{14}$$

where the different terms and values are detailed in Table 1.

The size and shape of the over-oxygenation is uncertain, as are the possible variations in oxygen sources and losses during
this process. We then arbitrarily define a log-polynomial fit for the time evolution of the parameters $\alpha_i = e^{(a(t-t_0)^{2p}+b)}$ ($\alpha_i =$
$\alpha_N$ or $\alpha_\Psi$). When there is no deviation from the initial model the values of $\alpha_i$ are equal to 1. The variation of the $\alpha_i$ parameters
is triggered in a way that is consistent with the predictions of previous studies on the evolution of primary productivity and ice
ages.





| Terms | Description | Values |
|---|---|---|
| Atmospheric fluxes | | |
| $\Psi_{O_2}[CH_4]^{0,7}$ | Photochemical oxidation | Parametrization |
| $s[CH_4]$ | Atmospheric escape | $s = 2.03 \times 10^{-5}\ yr^{-1}$ |
| Surface fluxes | | |
| $N$ | Oxygenic photosynthesis | $3.75 \times 10^{14}$ mol $O_2$ equiv.$yr^{-1}$ |
| $\Omega_{O_2}$ | Fraction of $O_2$ produced that reaches the atmosphere | $\Omega_{O_2} = (1-\gamma)(1-\delta)$ |
| $\gamma$ | Fraction consumed | $\gamma = [O_2]/(d_\gamma + [O_2])$ |
| | by heterotrophic respirers | $d_\gamma = 1.36 \times 10^{19}$ mol |
| $\delta$ | Fraction consumed | $\delta = [O_2]/(d_\delta + [O_2])$ |
| | by methanotrophs | $d_\delta = 2.73 \times 10^{17}$ mol |
| $w$ | Bulk organic carbon weathering rate | $6 \times 10^{-9}\ yr^{-1}$ |
| $\beta$ | Fraction of organic carbon burial | $2.66 \times 10^{-3}$ |
| $r$ | Ferrous iron reducing material | $F_V + F_M + F_W - F_B$ |
| | Anoxygenic photosynthesis | mol $O_2$ equiv.$yr^{-1}$ |
| $F_V$ | Volcanic flux of reductants | $1.59 \times 10^{10} \left(\frac{3.586}{3.586-t}\right)^{0.17}$ |
| $F_M$ | Metamorphic outgassing of reductants | $6.12 \times 10^{11} \left(\frac{4.11}{4.11-t}\right)^{0.7}$ |
| $F_B$ | Burial | $1.06 \times 10^{12} \left(\frac{3.653}{3.653-t}\right)^{0.2}$ |
| $F_W$ | Oxidative weathering | $3.7 \times 10^4 [O_2]^{0.4}$ |

**Table 1.** Equation dependencies and values from Goldblatt et al. (2006). Reductant model from Claire et al. (2006) with time $t$ [Gyrs] and amount of oxygen $[O_2]$ [mol].

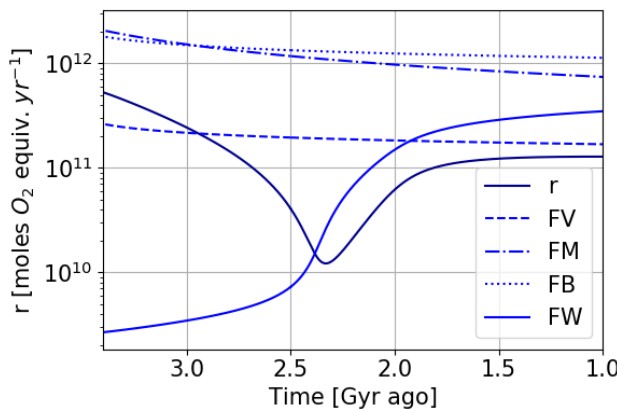

**Figure 13.** Temporal evolution of the reductant contributions (see Table 1) with $\alpha_i = 1$ and using for oxygen atmospheric loss ($F_{O_2}$) the Goldblatt et al. (2006) parametrization.





## 4.2 Dynamics with constant temperature and constant primary productivity

The Goldblatt et al. (2006) parameterization for atmospheric oxygen loss as a function of oxygen and methane abundance is compared (Figure 14) to the results obtained with the 1D GCM, which are shown to be identical to the 3D model. The parameterization established by Goldblatt et al. (2006) does not seem to correctly capture the methane dependence. This variation is not independent of oxygen abundance and cannot be described by a constant power $x$ of methane abundance $[CH_4]^x$. At high oxygen abundances ($> 10^{-4}$) the variation seems to increase with increasing methane, and conversely at lower oxygen abundances ($< 10^{-4}$). An asymptote appears to emerge at low oxygen abundances for the highest methane abundances ($> 10^{-5}$).

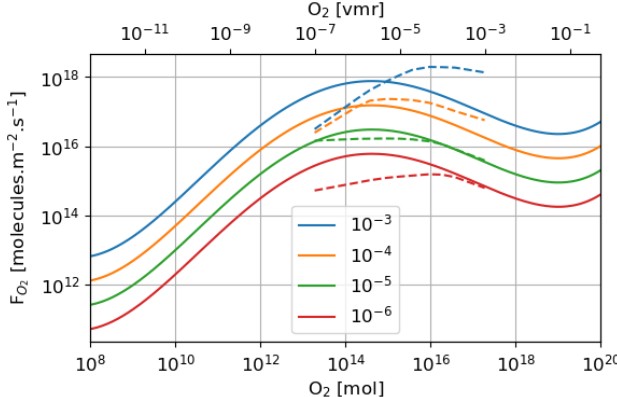

**Figure 14.** Oxygen atmospheric loss ($F_{O_2}$) depending on oxygen and methane (label in [vmr]). Goldblatt et al. (2006) parametrization (solid) and GCM 1D results (dash).

Figure 15 shows the equilibrium states as a function of the total reductant parameter $r$ with the Goldblatt et al. (2006) parameterization and an interpolation of the GCM results for the atmospheric losses. These curves show the stable and unstable equilibrium states of the atmosphere. They justify the rapid switch from an oxygen-poor to an oxygen-rich state. The flux of reductant that triggers the instability varies from about $1. \times 10^{10}$ to $3. \times 10^{10}$ mol O$_2$ equiv. yr$^{-1}$ with the interpolation of the GCM results.

A time evolution of oxygen and methane abundance with these two atmospheric loss models (Figure 16) shows that with the interpolation of GCM results oxygenation is faster. The equilibrium positions Figure 15 show that indeed the amount of oxygen is more sensitive to the flux of reductant. Oxygenation therefore occurs for a smaller variation of the reductant flux. As the methane abundance is directly related to the reductant flux ($[CH_4] = \frac{r}{s}$), we also observe a smaller variation of the methane abundance during oxygenation.

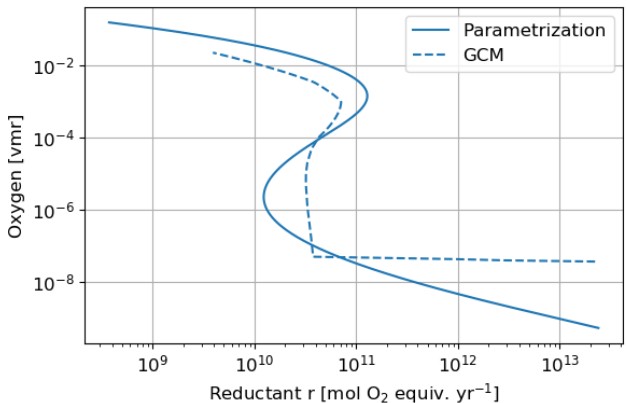

**Figure 15.** Equilibrium states of the time evolution equation model depending on the reductant parameter $r$. Oxygen atmospheric loss ($F_{O_2}$) Goldblatt et al. (2006) parametrization (solid) and GCM 1D results (dash).

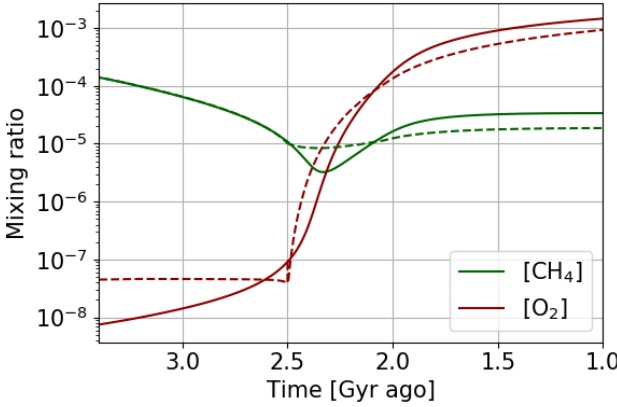

**Figure 16.** Temporal evolution of oxygen and methane with $\alpha_i = 1$. Oxygen atmospheric loss ($F_{O_2}$) Goldblatt et al. (2006) parametrization (solid) and GCM 1D results (dash).

### 4.3 Long overshoot with variable primary productivity

We model a long variation over 400 million years of the primary productivity, parameter $\alpha_N$, with a first phase of over-

production before a return to the initial production. We represent Figure 17 the reference evolution for constant $\alpha_N$ equal to 1, as well as two different intensities of over-productivity with a maximum at 2 and 10 times the initial productivity. The reference model is in good agreement with the results of Goldblatt et al. (2006) and Claire et al. (2006). We observe a variable over-oxygenation depending on the intensity of the primary productivity variation, but also an over-abundance of methane. The primary productivity corresponds to a photosynthetic production of oxygen but also of organic matter, transformed then

into methane by the methanogens. Consequently, the production of methane is increased as well as that of oxygen. Such an





over-abundance of methane is not highlighted by previous studies, nor by the geological record. It is difficult to constrain the amount of methane at that time. This scenario is therefore possible, although one could also imagine that oxygen enrichment of the atmosphere limited the conversion of organic matter into methane. A conversion that is carried out by methanogens, developing more favorably in a reducing environment. This negative feedback is not taken into account in the biosphere model.

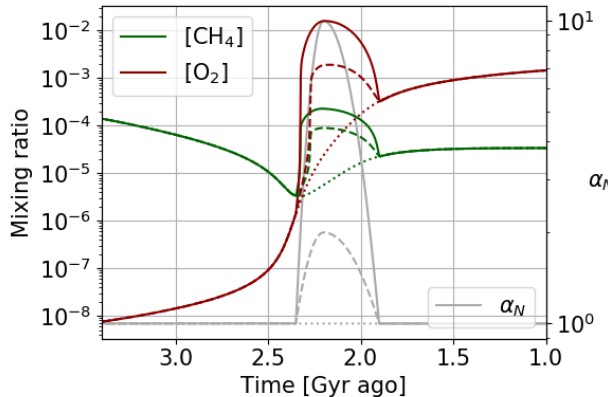

**Figure 17.** Oxygen and methane temporal evolution. Models $\alpha_N$ constant equal 1, reaching a factor 2 and reaching a factor 10.

This temporal evolution is identical if we model an inverse variation of atmospheric losses ($\alpha_\Psi = \frac{1}{\alpha_N}$), corresponding to a glaciation event of 400 million years. This scenario is less likely since there are several shorter glaciation events, called Huronian glaciations (Young et al., 2001). In addition, a change in primary productivity provides a link to the positive anomaly in carbon-13 isotope fractionation (Lyons et al., 2014). By using the model of the evolution of the isotopic ratio of Goldblatt et al. (2006), we can evaluate the impact of the variation of primary productivity or atmospheric losses on this ratio:

$$f = \frac{\delta_{carb} - \delta_i}{\delta_{carb} - \delta_{org}} \qquad (15)$$

where $f$ is the fraction of buried volcanic carbon, $\delta_{carb}$ is the carbon-13 isotope ratio $\delta^{13}C$ for carbonates, $\delta_i$ is the carbon-13 isotope ratio $\delta^{13}C$ for volcanic carbon, and $\delta_{org}$ is the carbon-13 isotope ratio $\delta^{13}C$ for organic carbon. The fraction of buried volcanic carbon can be described by the biosphere model based on the fraction C of buried carbon where $C \sim \frac{\beta}{w}(N + r)$. We establish the approximation that $f \alpha (N + r)$. Goldblatt et al. (2006) defines the initial state during the Archean with $\delta_{carb} = 0$

310 ‰, $\delta_i$ = -6 ‰ and $\delta_{org}$ = -30 ‰ giving $f$ = 0.2. With this initial state, we determine the relation of proportionality between $f$ and $(N + r)$, then we evaluate the evolution of the isotopic fractionation of the carbonates $\delta_{carb}$ during the temporal evolution of the oxygen and methane abundance thanks to the formula:

$$\delta_{carb} = \frac{\delta_i - f \times \delta_{org}}{1 - f} \qquad (16)$$





Figure 18 represents the evolution of $\delta_{carb}$ as a function of time for the time evolution models established with a primary
over-productivity reaching a factor of 2 and the inverse evolution of atmospheric losses. A primary over-productivity is consistent with the anomaly of about 15 ‰ measured in contrast to the decrease in atmospheric losses.

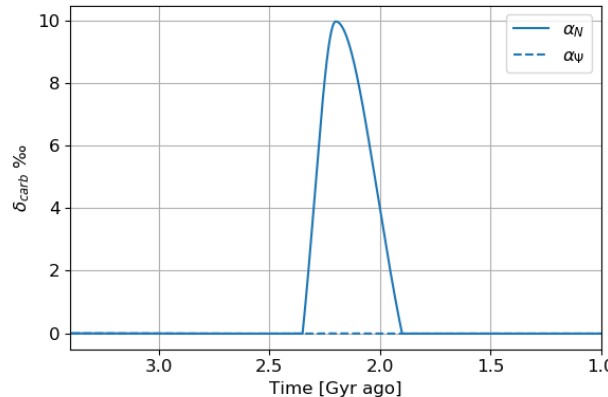

**Figure 18.** Temporal evolution of $\delta_{carb}$. Models with $\alpha_N$ reaching a factor 2 and $\alpha_\Psi$ reaching a factor 0.5.

## 4.4 Short overshoot with variable temperature

The Huronian glaciations represent several glacial events that took place during the oxygenation of the atmosphere between the
Archean and the Proterozoic. The model of the $\alpha_\Psi$ parameter variation over 400 million years is therefore not consistent with
the episodic nature of these glaciations. To reflect the impact of these glaciations on atmospheric losses, an episodic variation
over shorter times during the oxygenation period can be established. Figure 19 presents the temporal evolution of oxygen and
methane following this adjustment. We observe the punctual over-oxygenations linked to the variations of $\alpha_\Psi$ with a global
trend that follows the temporal evolution of the initial model where $\alpha_i = 1$. We note, during the over-oxygenation, an increase
in the methane abundance. This increase in greenhouse gases can trigger the thawing of the surface. During the thawing, the
surface temperature increases causing an increase in atmospheric losses by oxidation of methane. This is why the amount of
methane decreases again. This negative feedback, coupled with the hysteresis phenomenon between the frozen and warm state
of the surface, could be one of the key factors to explain the cyclic character of Huronian glaciations.

Figure 20 shows the complete model for over-oxygenation with a 400 million year variation in primary productivity and a
maximum factor of 2, as well as a fluctuation provided by shorter time scale variations in atmospheric losses due to Huronian
glaciations with a maximum factor of 0.5.

## 5 Conclusions

The atmospheric equilibrium states during the Archean and the Proterozoic has been establish for the first time using a 3D
photochemical-climate model. Despite some 3D transport discrepancies the atmospheric-surface equilibrium fluxes of methane

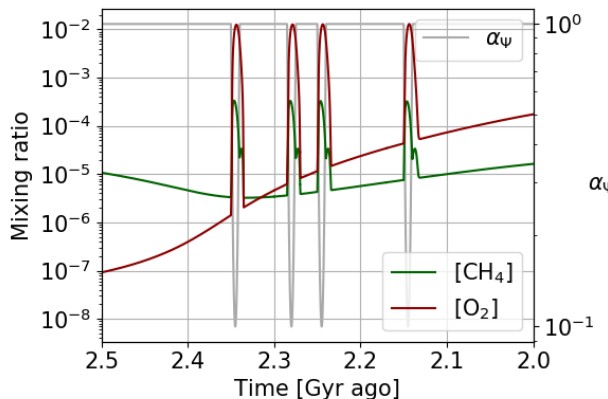

**Figure 19.** Oxygen and methane temporal evolution. Models $\alpha_\Psi$ reaching a factor 0.1 on shorter timescales.

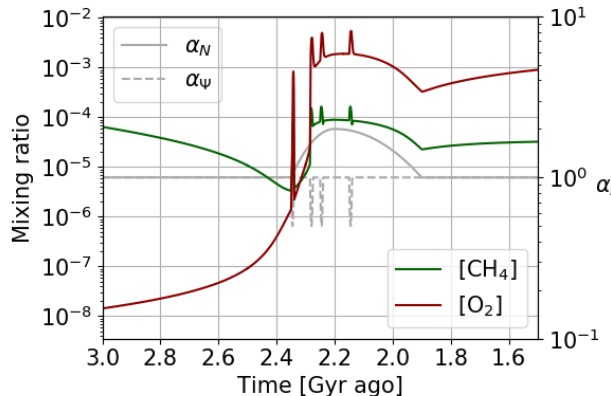

**Figure 20.** Oxygen and methane temporal evolution. Models $\alpha_N$ reaching a factor 2. Models $\alpha_\Psi$ reaching a factor 0.5 on shorter timescales.

and oxygen are not significantly different from a 1D model as it has been done in Zahnle et al. (2006) or Gebauer et al. (2017).
Following, the (photo)chemical equilibrium pathways have been determined depending on the altitude. It highlights the evolution of the tropospheric and stratospheric contribution depending of the oxygen levels. What remains however constant is the important link with the OH molecules which catalyze the methane oxidation. Because of that, we found a crucial dependency of the surface fluxes equilibrium with the surface temperature. This temperature dependence is sensitive to the 3D transport and appears weaker in 3D than in 1D. However, a global glaciation could reduce the oxygen atmospheric losses by a factor 2 to 10.
Taking this new contribution into account in a time evolution model, we show that glaciations bring fluctuations in oxygen and methane abundance with an overshoot during glaciations. The increase in methane following glaciation produces an additional greenhouse effect that could eventually lead to deglaciation. This warming increases again the atmospheric losses of methane and it is possible to establish a cycle of glaciation-deglaciation. Few evidences of methane (Lowe and Tice, 2004; Cadeau et al.,





2020) have been yet discovered but it could explain how the planet has been warmed up enough to terminate the glaciation
thanks to the methane greenhouse effect. The feedback between the glaciation and the methane evolution coupled with the glaciation hysteresis process might also explain the multiple glaciations known as the Huronian glaciations. More generally, the link between temperature and photochemical processes shows that a decrease in temperature favors the oxygenation of the atmosphere. Without falling into a global glaciation, phenomena such as the decrease of the atmospheric $CO_2$ or the emergence of continents induce a decrease of temperature favorable to the oxygenation of the atmosphere.

Beyond the temperature, the UV-Visible stellar radiation received by the planet controls the photochemical processes and the quantity of ozone, essential for the oxygenation phenomenon of the atmosphere. A red dwarf such as TRAPPIST-1a presents a spectral distribution in favor of the UV radiation with respect to the visible radiation which favors the accumulation of ozone. The 1D study of Gebauer et al. (2018) shows that around such red dwarfs the oxygenation of the atmosphere is triggered for lower atmospheric oxygen levels and surface oxygen flux. These irradiation conditions then favor the oxygenation of the
atmosphere. A compact system such as TRAPPIST-1 has presumably synchronous planets (Vinson and Hansen, 2017), which brings an interest to use 3D models. The permanent dichotomy between day and night side brings an important variation of temperature (Leconte et al., 2013) and photochemical processes. The red dwarfs represent the majority of the stars and would thus shelter the majority of the planets. It is then necessary to understand how an oxygenated atmosphere could evolve around these planets. For this, 3D models will be necessary to capture the effect of a synchronization.





**Appendix A: Chemical network**

**Table A1.** Chemical network with rate coefficients and references used to model the early Earth.

Units are $s^{-1}$ for photolysis reactions, $cm^3.s^{-1}$ for two-body reactions, and $cm^6.s^{-1}$ for three-body reactions. [M] correspond to the density in molecules.$cm^{-3}$. If $k_0$ and $k_\infty$ specified, the formula is $\frac{k_0[M]}{1+\frac{k_0}{k_\infty}[M]}0.6^{[1+[log_{10}(\frac{k_0}{k_\infty}[M])]^2]^{-1}}$. If $k_{1a,0}$, $k_{1b,0}$, $k_{1a,\infty}$ and $F_c$ specified, the

formula is $k_{1a,0}\frac{1+\frac{k_{1b,0}}{k_{1a,\infty}-k_{1b,0}}}{1+\frac{k_{1a,0}}{k_{1a,\infty}-k_{1b,0}}}F_c^{[1+[log(\frac{k_{1a,0}}{k_{1a,\infty}-k_{1b,0}})]^2]^{-1}} + k_{1b,0}\frac{1}{1+\frac{k_{1a,0}}{k_{1a,\infty}-k_{1b,0}}}F_c^{[1+[log(\frac{k_{1a,0}}{k_{1a,\infty}-k_{1b,0}})]^2]^{-1}}$

| Reaction | | | | | | | | Rate Coefficient | Reference |
|---|---|---|---|---|---|---|---|---|---|
| $O_2$ | + | $h\nu$ | $\longrightarrow$ | O | + | O | | $J_{O_2 \longrightarrow O}$ | Ogawa and Ogawa (1975) |
| $O_2$ | + | $h\nu$ | $\longrightarrow$ | O | + | $O(^1D)$ | | $J_{O_2 \longrightarrow O(^1D)}$ | and Lewis et al. (1983) |
| | | | | | | | | | and Gibson et al. (1983) |
| | | | | | | | | | and Minschwaner et al. (1992) |
| | | | | | | | | | and Yoshino et al. (1988) |
| | | | | | | | | | and Fally et al. (2000) |
| $CO_2$ | + | $h\nu$ | $\longrightarrow$ | CO | + | O | | $J_{CO_2 \longrightarrow O}$ | Chan et al. (1993) |
| $CO_2$ | + | $h\nu$ | $\longrightarrow$ | CO | + | $O(^1D)$ | | $J_{CO_2 \longrightarrow O(^1D)}$ | and Stark et al. (2007) |
| | | | | | | | | | and Yoshino et al. (1996) |
| | | | | | | | | | and Parkinson et al. (2003) |
| | | | | | | | | | and Lewis and Carver (1983) |
| $O_3$ | + | $h\nu$ | $\longrightarrow$ | $O_2$ | + | O | | $J_{O_3 \longrightarrow O}$ | Sander et al. (2006) |
| $O_3$ | + | $h\nu$ | $\longrightarrow$ | $O_2$ | + | $O(^1D)$ | | $J_{O_3 \longrightarrow O(^1D)}$ | |
| $H_2O$ | + | $h\nu$ | $\longrightarrow$ | OH | + | H | | $J_{H_2O}$ | Mota et al. (2005) |
| | | | | | | | | | and Chung et al. (2001) |
| | | | | | | | | | and Thompson et al. (1963) |
| $H_2O_2$ | + | $h\nu$ | $\longrightarrow$ | OH | + | OH | | $J_{H_2O_2}$ | Schurgers and Welge (1968) |
| | | | | | | | | | and Demore et al. (1997) |
| $HO_2$ | + | $h\nu$ | $\longrightarrow$ | OH | + | O | | $J_{HO_2}$ | Sander et al. (2003) |
| $CH_4$ | + | $h\nu$ | $\longrightarrow$ | $CH_3$ | + | H | | $J_{CH_4 \longrightarrow CH_3}$ | Kameta et al. (2002) |
| $CH_4$ | + | $h\nu$ | $\longrightarrow$ | $^1CH_2$ | + | $H_2$ | | $J_{CH_4 \longrightarrow {}^1CH_2}$ | and Chen and Wu (2004) |
| $CH_4$ | + | $h\nu$ | $\longrightarrow$ | $^3CH_2$ | + | H | + H | $J_{CH_4 \longrightarrow {}^3CH_2}$ | and Lee et al. (2001) |
| $CH_4$ | + | $h\nu$ | $\longrightarrow$ | CH | + | $H_2$ | + H | $J_{CH_4 \longrightarrow CH}$ | |
| $CH_2O$ | + | $h\nu$ | $\longrightarrow$ | CHO | + | H | | $J_{CH_2O \longrightarrow CHO}$ | Sander et al. (2011) |
| $CH_2O$ | + | $h\nu$ | $\longrightarrow$ | CO | + | $H_2$ | | $J_{CH_2O \longrightarrow CO}$ | |



| Reaction | | | | | | | Rate Coefficient | Reference |
|---|---|---|---|---|---|---|---|---|
| $C_2H_6$ | + | $h\nu$ | $\longrightarrow$ | $CH_4$ | + | $^1CH_2$ | $J_{C_2H_6 \longrightarrow CH_4}$ | Lee et al. (2001) |
| $C_2H_6$ | + | $h\nu$ | $\longrightarrow$ | $C_2H_2$ | + | $H_2$ + $H_2$ | $J_{C_2H_6 \longrightarrow C_2H_2}$ | |
| $C_2H_6$ | + | $h\nu$ | $\longrightarrow$ | $C_2H_4$ | + | $H$ + $H$ | $J_{C_2H_6 \longrightarrow C_2H_4+H}$ | |
| $C_2H_6$ | + | $h\nu$ | $\longrightarrow$ | $C_2H_4$ | + | $H_2$ | $J_{C_2H_6 \longrightarrow C_2H_4+H_2}$ | |
| $C_2H_6$ | + | $h\nu$ | $\longrightarrow$ | $CH_3$ | + | $CH_3$ | $J_{C_2H_6 \longrightarrow CH_3}$ | |
| $C_2H_4$ | + | $h\nu$ | $\longrightarrow$ | $C_2H_2$ | + | $H_2$ | $J_{C_2H_4 \longrightarrow H_2}$ | Kasting et al. (1983) |
| $C_2H_4$ | + | $h\nu$ | $\longrightarrow$ | $C_2H_2$ | + | $H$ + $H$ | $J_{C_2H_4 \longrightarrow H}$ | |
| $C_2H_2$ | + | $h\nu$ | $\longrightarrow$ | $C_2H$ | + | $H$ | $J_{C_2H_2 \longrightarrow C_2H}$ | Chen et al. (1991) |
| $C_2H_2$ | + | $h\nu$ | $\longrightarrow$ | $C_2$ | + | $H_2$ | $J_{C_2H_2 \longrightarrow C_2}$ | and Smith et al. (1991) |
| $CH_2CO$ | + | $h\nu$ | $\longrightarrow$ | $^3CH_2$ | + | $CO$ | $J_{CH_2CO}$ | Laufer and Keller (1971) |
| $O(^1D)$ | | | $\xrightarrow{CO_2}$ | $O$ | | | $7.5 \times 10^{-11} e^{\frac{115}{T}}$ | Sander et al. (2006) |
| $O(^1D)$ | | | $\xrightarrow{O_2}$ | $O$ | | | $3.3 \times 10^{-11} e^{\frac{55}{T}}$ | Sander et al. (2006) |
| $O(^1D)$ | | | $\xrightarrow{N_2}$ | $O$ | | | $1.8 \times 10^{-11} e^{\frac{110}{T}}$ | Sander et al. (2006) |
| $^1CH_2$ | | | $\xrightarrow{M}$ | $^3CH_2$ | | | $8.8 \times 10^{-12}$ | Ashfold et al. (1981) |
| $O$ | + | $O$ | $\longrightarrow$ | $O_2$ | | | $2.365 \times 10^{-33} e^{\frac{485}{T}}[M]$ | Campbell and Gray (1973) |
| $OH$ | + | $OH$ | $\longrightarrow$ | $H_2O$ | + | $O$ | $1.8 \times 10^{-12}$ | Sander et al. (2006) |
| $OH$ | + | $OH$ | $\longrightarrow$ | $H_2O_2$ | | | $k_0 = 6.9 \times 10^{-31} \left(\frac{T}{300}\right)^{-1}$ | Sander et al. (2003) |
| | | | | | | | $k_\infty = 2.6 \times 10^{-11}$ | |
| $HO_2$ | + | $HO_2$ | $\longrightarrow$ | $H_2O_2$ | + | $O_2$ | $1.5 \times 10^{-12} e^{\frac{19}{T}}$ | Christensen et al. (2002) |
| $HO_2$ | + | $HO_2$ | $\longrightarrow$ | $H_2O_2$ | + | $O_2$ | $2.1 \times 10^{-33} e^{\frac{920}{T}}[M]$ | Sander et al. (2011) |
| $H$ | + | $H$ | $\longrightarrow$ | $H_2$ | | | $1.8 \times 10^{-30} \frac{1}{T}[M]$ | Baulch et al. (2005) |
| $^3CH_2$ | + | $^3CH_2$ | $\longrightarrow$ | $C_2H_2$ | + | $H_2$ | $5.3 \times 10^{-11}$ | Banyard et al. (1980) and Laufer (1981) |
| $C_2H_3$ | + | $C_2H_3$ | $\longrightarrow$ | $C_2H_4$ | + | $C_2H_2$ | $2.4 \times 10^{-11}$ | Fahr et al. (1991) |
| $CHO$ | + | $CHO$ | $\longrightarrow$ | $CH_2O$ | + | $CO$ | $4.5 \times 10^{-11}$ | Friedrichs et al. (2002) |
| $CH_3$ | + | $CH_3$ | $\longrightarrow$ | $C_2H_6$ | | | $k_0 = 1.17 \times 10^{-25} \left(\frac{T}{300}\right)^{-3.75} e^{\frac{-500}{T}}$ | Wagner and Wardlaw (1988) |
| | | | | | | | $k_\infty = 3.0 \times 10^{-11} \left(\frac{T}{300}\right)^{-1}$ | and Wang et al. (2003) |
| $O$ | + | $O_2$ | $\longrightarrow$ | $O_3$ | | | $1.245 \times 10^{-33} \left(\frac{T}{300}\right)^{-2.4}[M]$ | Sander et al. (2003) |
| $O$ | + | $O_3$ | $\longrightarrow$ | $O_2$ | + | $O_2$ | $8.0 \times 10^{-12} e^{\frac{-2060}{T}}$ | Sander et al. (2003) |
| $O(^1D)$ | + | $H_2O$ | $\longrightarrow$ | $OH$ | + | $OH$ | $1.63 \times 10^{-10} e^{\frac{60}{T}}$ | Sander et al. (2006) |
| $O(^1D)$ | + | $H_2$ | $\longrightarrow$ | $OH$ | + | $H$ | $1.2 \times 10^{-10}$ | Sander et al. (2011) |
| $O(^1D)$ | + | $O_3$ | $\longrightarrow$ | $O_2$ | + | $O_2$ | $1.2 \times 10^{-10}$ | Sander et al. (2003) |
| $O(^1D)$ | + | $O_3$ | $\longrightarrow$ | $O_2$ | + | $O$ + $O$ | $1.2 \times 10^{-10}$ | Sander et al. (2003) |
| $O(^1D)$ | + | $CH_4$ | $\longrightarrow$ | $CH_3$ | + | $OH$ | $1.125 \times 10^{-10}$ | Sander et al. (2003) |
| $O(^1D)$ | + | $CH_4$ | $\longrightarrow$ | $CH_3O$ | + | $H$ | $3.0 \times 10^{-11}$ | Sander et al. (2003) |
| $O(^1D)$ | + | $CH_4$ | $\longrightarrow$ | $CH_2O$ | + | $H_2$ | $7.5 \times 10^{-12}$ | Sander et al. (2003) |
| $O$ | + | $HO_2$ | $\longrightarrow$ | $OH$ | + | $O_2$ | $3.0 \times 10^{-11} e^{\frac{200}{T}}$ | Sander et al. (2003) |
| $O$ | + | $OH$ | $\longrightarrow$ | $O_2$ | + | $H$ | $1.8 \times 10^{-11} e^{\frac{180}{T}}$ | Sander et al. (2011) |
| $H$ | + | $O_3$ | $\longrightarrow$ | $OH$ | + | $O_2$ | $1.4 \times 10^{-10} e^{\frac{-470}{T}}$ | Sander et al. (2003) |
| $H$ | + | $HO_2$ | $\longrightarrow$ | $OH$ | + | $OH$ | $7.2 \times 10^{-11}$ | Sander et al. (2006) |
| $H$ | + | $HO_2$ | $\longrightarrow$ | $H_2$ | + | $O_2$ | $6.9 \times 10^{-12}$ | Sander et al. (2006) |
| $H$ | + | $HO_2$ | $\longrightarrow$ | $H_2O$ | + | $O$ | $1.6 \times 10^{-12}$ | Sander et al. (2006) |



| | Reaction | | | | | | | Rate Coefficient | Reference |
|---|---|---|---|---|---|---|---|---|---|
| OH | + | HO$_2$ | $\longrightarrow$ | H$_2$O | + | O$_2$ | | $4.8 \times 10^{-11} e^{\frac{250}{T}}$ | Sander et al. (2003) |
| OH | + | H$_2$O$_2$ | $\longrightarrow$ | H$_2$O | + | HO$_2$ | | $1.8 \times 10^{-12}$ | Sander et al. (2006) |
| OH | + | H$_2$ | $\longrightarrow$ | H$_2$O | + | H | | $2.8 \times 10^{-12} e^{\frac{-1800}{T}}$ | Sander et al. (2006) |
| H | + | O$_2$ | $\longrightarrow$ | HO$_2$ | | | | $k_0 = 4.4 \times 10^{-32} \left(\frac{T}{300}\right)^{-1.3}$ | Sander et al. (2011) |
| | | | | | | | | $k_\infty = 7.5 \times 10^{-11} \left(\frac{T}{300}\right)^{0.2}$ | |
| O | + | H$_2$O$_2$ | $\longrightarrow$ | OH | + | HO$_2$ | | $1.4 \times 10^{-12} e^{\frac{-2000}{T}}$ | Sander et al. (2003) |
| OH | + | O$_3$ | $\longrightarrow$ | HO$_2$ | + | O$_2$ | | $1.7 \times 10^{-12} e^{\frac{-940}{T}}$ | Sander et al. (2003) |
| HO$_2$ | + | O$_3$ | $\longrightarrow$ | OH | + | O$_2$ | + O$_2$ | $1.0 \times 10^{-14} e^{\frac{-490}{T}}$ | Sander et al. (2003) |
| CO | + | O | $\longrightarrow$ | CO$_2$ | | | | $1.625 \times 10^{-32} e^{\frac{-2184}{T}}$[M] | Tsang and Hampson (1986) |
| CO | + | OH | $\longrightarrow$ | CO$_2$ | + | H | | $k_{1a,0} = 1.34[M] \times 3.62 \times 10^{-26} T^{-2.739} e^{\frac{-20}{T}}$ | Joshi and Wang (2006) |
| | | | | | | | | $+ [6.48 \times 10^{-33} T^{0.14} e^{\frac{-57}{T}}]^{-1}$ | |
| | | | | | | | | $k_{1b,0} = 1.17 \times 10^{-19} T^{2.053} e^{\frac{139}{T}}$ | |
| | | | | | | | | $+ 9.56 \times 10^{-12} T^{-0.664} e^{\frac{-167}{T}}$ | |
| | | | | | | | | $k_{1a,\infty} = 1.52 \times 10^{-17} T^{1.858} e^{\frac{28.8}{T}}$ | |
| | | | | | | | | $+ 4.78 \times 10^{-8} T^{-1.851} e^{\frac{-318}{T}}$ | |
| | | | | | | | | $F_c = 0.628 e^{\frac{-1223}{T}} + (1 - 0.628) e^{\frac{-39}{T}} + e^{\frac{-T}{255}}$ | |
| C | + | H$_2$ | $\longrightarrow$ | $^3$CH$_2$ | | | | $\frac{8.75 \times 10^{-31} e^{\frac{524}{T}}[M]}{1 + \frac{8.75 \times 10^{-31} e^{\frac{524}{T}}}{8.3 \times 10^{-11}}[M]}$ | Zahnle (1986) |
| C | + | O$_2$ | $\longrightarrow$ | CO | + | O | | $3.3 \times 10^{-11}$ | Donovan and Husain (1970) |
| C | + | OH | $\longrightarrow$ | CO | + | H | | $4.0 \times 10^{-11}$ | Giguere and Huebner (1978) |
| C$_2$ | + | CH$_4$ | $\longrightarrow$ | C$_2$H | + | CH$_3$ | | $5.05 \times 10^{-11} e^{\frac{-297}{T}}$ | Pitts et al. (1982) |
| C$_2$ | + | H$_2$ | $\longrightarrow$ | C$_2$H | + | H | | $1.77 \times 10^{-10} e^{\frac{-1469}{T}}$ | Pitts et al. (1982) |
| C$_2$ | + | O | $\longrightarrow$ | C | + | CO | | $5.0 \times 10^{-11}$ | Prasad and Huntress (1980) |
| C$_2$ | + | O$_2$ | $\longrightarrow$ | CO | + | CO | | $1.5 \times 10^{-11} e^{\frac{-550}{T}}$ | Baughcum and Oldenborg (1983) |
| C$_2$H | + | C$_2$H$_2$ | $\longrightarrow$ | Hcaer | + | H | | $1.5 \times 10^{-10}$ | Stephens et al. (1988) |
| C$_2$H | + | CH$_4$ | $\longrightarrow$ | C$_2$H$_2$ | + | CH$_3$ | | $6.94 \times 10^{-12} e^{\frac{-250}{T}}$ | Lander et al. (1990) and Allen et al. (1992) |
| C$_2$H | + | H | $\longrightarrow$ | C$_2$H$_2$ | | | | $\frac{1.26 \times 10^{-18} T^{-3.1} e^{\frac{-721}{T}}[M]}{1 + \frac{1.26 \times 10^{-18} T^{-3.1} e^{\frac{-721}{T}}}{3.0 \times 10^{-10}}[M]}$ | Tsang and Hampson (1986) |
| C$_2$H | + | H$_2$ | $\longrightarrow$ | C$_2$H$_2$ | + | H | | $5.58 \times 10^{-11} e^{\frac{-1443}{T}}$ | Stephens et al. (1988) and Allen et al. (1992) |
| C$_2$H | + | O | $\longrightarrow$ | CO | + | CH | | $1.0 \times 10^{-10} e^{\frac{-250}{T}}$ | Zahnle (1986) |
| C$_2$H | + | O$_2$ | $\longrightarrow$ | CO | + | CHO | | $2.0 \times 10^{-11}$ | Brown and Laufer (1982) |
| C$_2$H$_2$ | + | H | $\longrightarrow$ | C$_2$H$_3$ | | | | $\frac{2.6 \times 10^{-31}[M]}{1 + \frac{2.6 \times 10^{-31}}{8.3 \times 10^{-11} e^{\frac{-1374}{T}}}[M]}$ | Romani et al. (1993) |
| C$_2$H$_2$ | + | O | $\longrightarrow$ | $^3$CH$_2$ | + | CO | | $2.9 \times 10^{-11} e^{\frac{-1600}{T}}$ | Zahnle (1986) |
| C$_2$H$_2$ | + | OH | $\longrightarrow$ | CH$_2$CO | + | H | | $\frac{5.8 \times 10^{-31} e^{\frac{1258}{T}}[M]}{1 + \frac{5.8 \times 10^{-31} e^{\frac{1258}{T}}}{1.4 \times 10^{-12} e^{\frac{388}{T}}}[M]}$ | Perry and Williamson (1982) |
| C$_2$H$_2$ | + | OH | $\longrightarrow$ | CO | + | CH$_3$ | | $2.0 \times 10^{-12} e^{\frac{-250}{T}}$ | Hampson and Garvin (1977) |
| C$_2$H$_3$ | + | CH$_3$ | $\longrightarrow$ | C$_2$H$_2$ | + | CH$_4$ | | $3.4 \times 10^{-11}$ | Fahr et al. (1991) |
| C$_2$H$_3$ | + | CH$_4$ | $\longrightarrow$ | C$_2$H$_4$ | + | CH$_3$ | | $2.4 \times 10^{-24} e^{\frac{-2754}{T}}$ | Tsang and Hampson (1986) |
| C$_2$H$_3$ | + | H | $\longrightarrow$ | C$_2$H$_2$ | + | H$_2$ | | $3.3 \times 10^{-11}$ | Warnatz (1984) |
| C$_2$H$_3$ | + | H$_2$ | $\longrightarrow$ | C$_2$H$_4$ | + | H | | $2.6 \times 10^{-13} e^{\frac{-2646}{T}}$ | Allen et al. (1992) |
| C$_2$H$_3$ | + | O | $\longrightarrow$ | CH$_2$CO | + | H | | $5.5 \times 10^{-11}$ | Hoyermann et al. (1981) |
| C$_2$H$_3$ | + | OH | $\longrightarrow$ | C$_2$H$_2$ | + | H$_2$O | | $8.3 \times 10^{-12}$ | Benson and Haugen (1967) |
| C$_2$H$_4$ | + | O | $\longrightarrow$ | CHO | + | CH$_3$ | | $5.5 \times 10^{-12} e^{\frac{-565}{T}}$ | Hampson and Garvin (1977) |
| C$_2$H$_4$ | + | OH | $\longrightarrow$ | CH$_2$O | + | CH$_3$ | | $2.2 \times 10^{-12} e^{\frac{385}{T}}$ | Hampson and Garvin (1977) |



| Reaction | | | | | | | | Rate Coefficient | Reference |
|---|---|---|---|---|---|---|---|---|---|
| CH | + | $CO_2$ | $\longrightarrow$ | CHO | + | CO | | $5.9 \times 10^{-12} e^{\frac{-350}{T}}$ | Berman et al. (1982) |
| CH | + | H | $\longrightarrow$ | C | + | $H_2$ | | $1.4 \times 10^{-11}$ | Becker et al. (1989) |
| CH | + | $H_2$ | $\longrightarrow$ | $^3CH_2$ | + | H | | $2.38 \times 10^{-10} e^{\frac{-1760}{T}}$ | Zabarnick et al. (1986) |
| CH | + | $H_2$ | $\longrightarrow$ | $CH_3$ | | | | $\dfrac{8.75 \times 10^{-31} e^{\frac{524}{T}} [M]}{1 + \frac{8.75 \times 10^{-31} e^{\frac{524}{T}}}{8.3 \times 10^{-11}} [M]}$ | Romani et al. (1993) |
| CH | + | O | $\longrightarrow$ | CO | + | H | | $9.5 \times 10^{-11}$ | Messing et al. (1981) |
| CH | + | $O_2$ | $\longrightarrow$ | CO | + | OH | | $5.9 \times 10^{-11}$ | Butler et al. (1981) |
| CH | + | $CH_4$ | $\longrightarrow$ | $C_2H_4$ | + | H | | $\min(2.5 \times 10^{-11} e^{\frac{200}{T}}, 1.7 \times 10^{-10})$ | Romani et al. (1993) |
| $^1CH_2$ | + | $CH_4$ | $\longrightarrow$ | $CH_3$ | + | $CH_3$ | | $7.14 \times 10^{-12} e^{\frac{-5050}{T}}$ | Böhland et al. (1985) |
| $^1CH_2$ | + | $CO_2$ | $\longrightarrow$ | $CH_2O$ | + | CO | | $1.0 \times 10^{-12}$ | Zahnle (1986) |
| $^1CH_2$ | + | $H_2$ | $\longrightarrow$ | $^3CH_2$ | + | $H_2$ | | $1.26 \times 10^{-11}$ | Romani et al. (1993) |
| $^1CH_2$ | + | $H_2$ | $\longrightarrow$ | $CH_3$ | + | H | | $5.0 \times 10^{-15}$ | Tsang and Hampson (1986) |
| $^1CH_2$ | + | $O_2$ | $\longrightarrow$ | CHO | + | OH | | $3.0 \times 10^{-11}$ | Ashfold et al. (1981) |
| $^3CH_2$ | + | $C_2H_3$ | $\longrightarrow$ | $CH_3$ | + | $C_2H_2$ | | $3.0 \times 10^{-11}$ | Tsang and Hampson (1986) |
| $^3CH_2$ | + | $CH_3$ | $\longrightarrow$ | $C_2H_4$ | + | H | | $7.0 \times 10^{-11}$ | Tsang and Hampson (1986) |
| $^3CH_2$ | + | CO | $\longrightarrow$ | $CH_2CO$ | | | | $\dfrac{1.0 \times 10^{-28} [M]}{1 + \frac{1.0 \times 10^{-28}}{1.0 \times 10^{-15}} [M]}$ | Yung et al. (1984) |
| $^3CH_2$ | + | $CO_2$ | $\longrightarrow$ | $CH_2O$ | + | CO | | $1.0 \times 10^{-14}$ | Darwin and Moore (1995) |
| $^3CH_2$ | + | H | $\longrightarrow$ | CH | + | $H_2$ | | $4.7 \times 10^{-10} e^{\frac{-370}{T}}$ | Zabarnick et al. (1986) |
| $^3CH_2$ | + | H | $\longrightarrow$ | $CH_3$ | | | | $\dfrac{3.1 \times 10^{-30} e^{\frac{475}{T}} [M]}{1 + \frac{3.1 \times 10^{-30} e^{\frac{475}{T}}}{1.5 \times 10^{-10}} [M]}$ | Gladstone (1983) |
| $^3CH_2$ | + | O | $\longrightarrow$ | CH | + | OH | | $8.0 \times 10^{-12}$ | Huebner and Giguere (1980) |
| $^3CH_2$ | + | O | $\longrightarrow$ | CO | + | H | + | H | $8.3 \times 10^{-11}$ | Homann and Schweinfurth (1981) |
| $^3CH_2$ | + | O | $\longrightarrow$ | CHO | + | H | | $1.0 \times 10^{-11}$ | Huebner and Giguere (1980) |
| $^3CH_2$ | + | $O_2$ | $\longrightarrow$ | CHO | + | OH | | $4.1 \times 10^{-11} e^{\frac{-750}{T}}$ | Baulch et al. (1994) |
| $^3CH_2$ | + | $C_2H_3$ | $\longrightarrow$ | $CH_3$ | + | $C_2H_2$ | | $3.0 \times 10^{-11}$ | Tsang and Hampson (1986) |
| $CH_2CO$ | + | H | $\longrightarrow$ | $CH_3$ | + | CO | | $1.9 \times 10^{-11} e^{\frac{-1725}{T}}$ | Michael et al. (1979) |
| $CH_2CO$ | + | O | $\longrightarrow$ | $CH_2O$ | + | CO | | $3.3 \times 10^{-11}$ | Lee (1980) and Miller et al. (1982) |
| $CH_3$ | + | CO | $\longrightarrow$ | $CH_3CO$ | | | | $1.4 \times 10^{-32} e^{\frac{-3000}{T}} [M]$ | Watkins and Word (1974) |
| $CH_3$ | + | H | $\longrightarrow$ | $CH_4$ | | | | $k_0 = 1.0 \times 10^{-28} (\frac{T}{300})^{-1.8}$ $k_\infty = 2.0 \times 10^{-10} (\frac{T}{300})^{-0.4}$ | Baulch et al. (1994) |
| $CH_3$ | + | $CH_2O$ | $\longrightarrow$ | $CH_4$ | + | CHO | | $1.6 \times 10^{-16} (\frac{T}{298})^{6.1} e^{\frac{899}{T}}$ | Baulch et al. (1994) |
| $CH_3$ | + | CHO | $\longrightarrow$ | $CH_4$ | + | CO | | $5.0 \times 10^{-11}$ | Tsang and Hampson (1986) |
| $CH_3$ | + | O | $\longrightarrow$ | $CH_2O$ | + | H | | $1.1 \times 10^{-10}$ | Sander et al. (2006) |
| $CH_3$ | + | $O_2$ | $\longrightarrow$ | $CH_2O$ | + | OH | | $k_0 = 4.5 \times 10^{-28} (\frac{T}{300})^{-3.0}$ $k_\infty = 1.8 \times 10^{-12} (\frac{T}{300})^{-1.7}$ | Sander et al. (2006) |
| $CH_3$ | + | $O_3$ | $\longrightarrow$ | $CH_2O$ | + | $HO_2$ | | $5.4 \times 10^{-12} e^{\frac{-220}{T}}$ | Sander et al. (2006) |
| $CH_3$ | + | $O_3$ | $\longrightarrow$ | $CH_3O$ | + | $O_2$ | | $5.4 \times 10^{-12} e^{\frac{-220}{T}}$ | Sander et al. (2006) |
| $CH_3$ | + | OH | $\longrightarrow$ | $CH_3O$ | + | H | | $9.3 \times 10^{-11} \frac{T}{298} e^{\frac{-1606}{T}}$ | Jasper et al. (2007) |
| $CH_3$ | + | OH | $\longrightarrow$ | CO | + | $H_2$ | + | $H_2$ | $6.7 \times 10^{-12}$ | Fenimore (1969) |

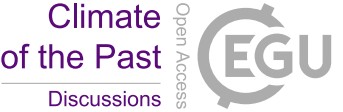

| Reaction | | | | | | Rate Coefficient | Reference |
|---|---|---|---|---|---|---|---|
| $CH_3CO$ | + | $CH_3$ | $\longrightarrow$ | $C_2H_6$ | + | CO | $5.4 \times 10^{-11}$ | Adachi et al. (1981) |
| $CH_3CO$ | + | $CH_3$ | $\longrightarrow$ | $CH_4$ | + | $CH_2CO$ | $8.6 \times 10^{-11}$ | Adachi et al. (1981) |
| $CH_3CO$ | + | H | $\longrightarrow$ | $CH_4$ | + | CO | $1.0 \times 10^{-10}$ | Zahnle (1986) |
| $CH_3CO$ | + | O | $\longrightarrow$ | $CH_2O$ | + | CHO | $5.0 \times 10^{-11}$ | Zahnle (1986) |
| $CH_3O$ | + | CO | $\longrightarrow$ | $CH_3$ | + | $CO_2$ | $2.6 \times 10^{-11}e^{\frac{-5940}{T}}$ | Wen et al. (1989) |
| $CH_4$ | + | O | $\longrightarrow$ | $CH_3$ | + | OH | $8.75 \times 10^{-12}\left(\frac{T}{298}\right)^{1.5}e^{\frac{-4330}{T}}$ | Tsang and Hampson (1986) |
| $CH_4$ | + | OH | $\longrightarrow$ | $CH_3$ | + | $H_2O$ | $2.45 \times 10^{-12}e^{\frac{-1775}{T}}$ | Sander et al. (2006) |
| H | + | CO | $\longrightarrow$ | CHO | | | $1.4 \times 10^{-34}e^{\frac{-1OO}{T}}$ [M] | Baulch et al. (1994) |
| H | + | CHO | $\longrightarrow$ | $H_2$ | + | CO | $1.8 \times 10^{-10}$ | Baulch et al. (1992) |
| $CH_2O$ | + | H | $\longrightarrow$ | $H_2$ | + | CHO | $2.14 \times 10^{-12}\left(\frac{T}{298}\right)^{1.62}e^{\frac{-1090}{T}}$ | Baulch et al. (1994) |
| $CH_2O$ | + | O | $\longrightarrow$ | CHO | + | OH | $3.4 \times 10^{-11}e^{\frac{-1600}{T}}$ | Sander et al. (2006) |
| $CH_2O$ | + | OH | $\longrightarrow$ | $H_2O$ | + | CHO | $5.5 \times 10^{-12}e^{\frac{125}{T}}$ | Sander et al. (2006) |
| CHO | + | $CH_2O$ | $\longrightarrow$ | $CH_3O$ | + | CO | $3.8 \times 10^{-17}$ | Wen et al. (1989) |
| CHO | + | $O_2$ | $\longrightarrow$ | $HO_2$ | + | CO | $5.2 \times 10^{-12}$ | Sander et al. (2006) |
| O | + | CHO | $\longrightarrow$ | H | + | $CO_2$ | $5.0 \times 10^{-11}$ | Tsang and Hampson (1986) |
| O | + | CHO | $\longrightarrow$ | OH | + | CO | $1.0 \times 10^{-10}$ | Hampson and Garvin (1977) |
| OH | + | CHO | $\longrightarrow$ | $H_2O$ | + | CO | $1.0 \times 10^{-10}$ | Tsang and Hampson (1986) |



## Appendix B:  Methane oxidation pathways

**Bilan :**

$$CH_4 + 2O_2 \longrightarrow CO_2 + 2H_2O \tag{B1}$$

**Tropospheric dominant pathways**

| | | | | | | |
|---|---|---|---|---|---|---|
| $CH_4$ | + | OH | $\longrightarrow$ | $CH_3$ | + | $H_2O$ |
| $CH_3$ | + | $O_2$ | $\longrightarrow$ | $CH_2O$ | + | OH |

| | | | | | | |
|---|---|---|---|---|---|---|
| $CH_2O$ | + | $h\nu$ | $\longrightarrow$ | CO | + | $H_2$ |
| $H_2$ | + | OH | $\longrightarrow$ | $H_2O$ | + | H |
| CO | + | OH | $\longrightarrow$ | $CO_2$ | + | H |
| H | + | $O_2$ | $\longrightarrow$ | $HO_2$ | | |
| H | + | $O_2$ | $\longrightarrow$ | $HO_2$ | | |
| $HO_2$ | + | $HO_2$ | $\longrightarrow$ | $H_2O_2$ | + | $O_2$ |
| $HO_2$ | + | $h\nu$ | $\longrightarrow$ | OH | + | OH |

| | | | | | | |
|---|---|---|---|---|---|---|
| $CH_2O$ | + | $h\nu$ | $\longrightarrow$ | CHO | + | H |
| CHO | + | $O_2$ | $\longrightarrow$ | CO | + | $HO_2$ |
| CO | + | OH | $\longrightarrow$ | $CO_2$ | + | H |
| H | + | $O_2$ | $\longrightarrow$ | $HO_2$ | | |
| H | + | $O_2$ | $\longrightarrow$ | $HO_2$ | | |
| $HO_2$ | + | $HO_2$ | $\longrightarrow$ | $H_2O_2$ | + | $O_2$ |
| $HO_2$ | + | $h\nu$ | $\longrightarrow$ | OH | + | OH |
| $HO_2$ | + | $HO_2$ | $\longrightarrow$ | $H_2O_2$ | + | $O_2$ |
| $H_2O_2$ | + | OH | $\longrightarrow$ | $H_2O$ | + | $HO_2$ |



## Stratospheric dominant pathway

$$CH_4 + OH \longrightarrow CH_3 + H_2O$$
$$CH_3 + O_2 \longrightarrow CH_2O + OH$$
$$CH_2O + OH \longrightarrow CHO + H_2O$$
$$CHO + O_2 \longrightarrow CO + HO_2$$
$$CO + OH \longrightarrow CO_2 + H$$
$$HO_2 + O \longrightarrow OH + O_2$$
$$H + O_2 \longrightarrow HO_2$$
$$HO_2 + h\nu \longrightarrow OH + O$$

## Upper atmosphere dominant pathway

$$CH_4 + h\nu \longrightarrow {}^1CH_2 + H_2$$
$${}^1CH_2 + O_2 \longrightarrow CHO + OH$$
$$CHO + O_2 \longrightarrow CO + HO_2$$
$$CO + OH \longrightarrow CO_2 + H$$
$$HO_2 + O \longrightarrow OH + O_2$$
$$H_2 + O(^1D) \longrightarrow OH + H$$
$$O_2 + h\nu \longrightarrow O + O(^1D)$$
$$H + O_2 \longrightarrow HO_2$$
$$H + O_2 \longrightarrow HO_2$$
$$HO_2 + OH \longrightarrow H_2O + O_2$$
$$HO_2 + OH \longrightarrow H_2O + O_2$$





**Upper atmosphere secondary pathways (lower half)**

$$CH_4 + h\nu \longrightarrow CH_3 + H$$
$$CH_3 + O_2 \longrightarrow CH_2O + OH$$

| | | | | | | | | | | | | |
|---|---|---|---|---|---|---|---|---|---|---|---|---|
| $CH_2O$ | + | $h\nu$ | $\longrightarrow$ | $CO$ | + | $H_2$ | | $CH_2O$ | + | $h\nu$ | $\longrightarrow$ | $CHO$ + $H$ |
| $HO_2$ | + | $O$ | $\longrightarrow$ | $OH$ | + | $O_2$ | | $CHO$ | + | $O_2$ | $\longrightarrow$ | $CO$ + $HO_2$ |
| $CO$ | + | $OH$ | $\longrightarrow$ | $CO_2$ | + | $H$ | | $CO$ | + | $OH$ | $\longrightarrow$ | $CO_2$ + $H$ |
| $H_2$ | + | $O(^1D)$ | $\longrightarrow$ | $OH$ | + | $H$ | | $HO_2$ | + | $H$ | $\longrightarrow$ | $H_2O$ + $O$ |
| $O_2$ | + | $h\nu$ | $\longrightarrow$ | $O$ | + | $O(^1D)$ | | $HO_2$ | + | $O$ | $\longrightarrow$ | $OH$ + $O_2$ |
| $H$ | + | $O_2$ | $\longrightarrow$ | $HO_2$ | | | | $H$ | + | $O_2$ | $\longrightarrow$ | $HO_2$ |
| $H$ | + | $O_2$ | $\longrightarrow$ | $HO_2$ | | | | $H$ | + | $O_2$ | $\longrightarrow$ | $HO_2$ |
| $H$ | + | $O_2$ | $\longrightarrow$ | $HO_2$ | | | | $HO_2$ | + | $OH$ | $\longrightarrow$ | $H_2O$ + $O_2$ |
| $HO_2$ | + | $OH$ | $\longrightarrow$ | $H_2O$ | + | $O_2$ | | | | | | |
| $HO_2$ | + | $OH$ | $\longrightarrow$ | $H_2O$ | + | $O_2$ | | | | | | |

**main**



$$CH_4 \;+\; h\nu \;\longrightarrow\; CH_3 \;+\; H$$
$$CH_3 \;+\; O \;\longrightarrow\; CH_2O \;+\; H$$

$\swarrow$                                   $\searrow$

| | | | | | |
|---|---|---|---|---|---|
| $CH_2O$ | + | $h\nu$ | $\longrightarrow$ $CO$ | + | $H_2$ |
| $HO_2$ | + | $O$ | $\longrightarrow$ $OH$ | + | $O_2$ |
| $CO$ | + | $OH$ | $\longrightarrow$ $CO_2$ | + | $H$ |
| $H_2$ | + | $O(^1D)$ | $\longrightarrow$ $OH$ | + | $H$ |
| $O_2$ | + | $h\nu$ | $\longrightarrow$ $O$ | + | $O(^1D)$ |
| $HO_2$ | + | $H$ | $\longrightarrow$ $H_2O$ | + | $O$ |
| $H$ | + | $O_2$ | $\longrightarrow$ $HO_2$ | | |
| $H$ | + | $O_2$ | $\longrightarrow$ $HO_2$ | | |
| $H$ | + | $O_2$ | $\longrightarrow$ $HO_2$ | | |
| $HO_2$ | + | $OH$ | $\longrightarrow$ $H_2O$ | + | $O_2$ |

| | | | | | |
|---|---|---|---|---|---|
| $CH_2O$ | + | $h\nu$ | $\longrightarrow$ $CHO$ | + | $H$ |
| $CHO$ | + | $O_2$ | $\longrightarrow$ $CO$ | + | $HO_2$ |
| $CO$ | + | $OH$ | $\longrightarrow$ $CO_2$ | + | $H$ |
| $HO_2$ | + | $O$ | $\longrightarrow$ $OH$ | + | $O_2$ |
| $HO_2$ | + | $H$ | $\longrightarrow$ $H_2O$ | + | $O$ |
| $HO_2$ | + | $H$ | $\longrightarrow$ $H_2O$ | + | $O$ |
| $H$ | + | $O_2$ | $\longrightarrow$ $HO_2$ | | |
| $H$ | + | $O_2$ | $\longrightarrow$ $HO_2$ | | |

**main**



*Author contributions.* Adam Yassin Jaziri participate to the conceptualization, investigation and methodology. Benjamin Charnay coordinate the porject and participate to the conceptualization, supervision and validation. Franck Selsis participate to the supervision and the validation. Jérémy Leconte provide the funding acquisition, resources, participate to the software check up and the validation. Franck Lefèvre and Adam Yassin Jaziri developped the part of the software needed for this work. Adam Yassin Jaziri made the visualization and write the original draft while Benjamin Charnay, Franck Selsis, Jérémy Leconte and Franck Lefèvre participate to the review and editing process.

*Competing interests.* The authors declare that they have no conflict of interest.

*Acknowledgements.* This project has received funding from the European Research Council (ERC) under the European Union's Horizon 2020 research and innovation programme (grant agreement No. 679030/WHIPLASH).




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
