# Peer review of "Dynamics of the Great Oxidation Event from a 3D photochemical-climate model"

_Climate of the Past, 2021_

## Author Response (AR1)

Clim. Past Discuss., author comment AC1
https://doi.org/10.5194/cp-2021-150-AC1, 2022

[Figure]

**Reply on RC1**

Adam Yassin Jaziri et al.

Author comment on "Dynamics of the Great Oxidation Event from a 3D photochemical-climate model" by Adam Yassin Jaziri et al., Clim. Past Discuss., https://doi.org/10.5194/cp-2021-150-AC1, 2022

This very constructive comment raises several misunderstanding. Mainly, the redox balance of the early atmosphere and the effect of surface temperature on the biosphere are discussed. We try with this answer to better clarify our approach which is in agreement with the comments.

In the atmospheric model, the redox balance of the early atmosphere is ensured by atmospheric chemistry and a surface flux that combines the different contributions mentioned ("organic carbon burial and the input of reductants"). The objective of the atmospheric model is to quantify the atmospheric contribution which establish a dynamic equilibrium with all the fluxes at the surface. This does not seem to be correctly underlined in the article. This is therefore the subject of a modification in part 2 "Model", aiming to clarify the meaning of the surface flux, corresponding mainly to the organic carbon burial and the input of reductants. Thereafter, all the contributions are well separated for the study of the temporal evolution.

We also agree that atmospheric instability did not "trigger" the GOE. The evolution of the organic carbon burial or other fluxes must be the cause. The atmospheric contribution is only at the origin of an instability that allows this rapid evolution from a low oxidized state to a high oxidized state, which control the oxygen levels without being the triggerer of the GOE. This also seems unclear in the article and may be subject to some linguistic adjustments.

These different misunderstanding are now better explained. Also, there seems to be confusion about Figure 15. The results of the 1D and 3D atmospheric simulations in terms of atmospheric loss flux are largely similar. Figure 15, as the legend indicates, then compares the result of 1D simulations with the theoretical model of Goldblatt et al. (2006). The interest in the decrease of the bistability zone does not come from the 3D model but only from atmospheric models (whose 1D approximation seems to be sufficient). Moreover, Figure 16 shows that despite this the dynamics remain similar. But it is true that we could insist more on this new result to emphasize the easier return from a high oxidized state to a low oxidized state.

We answer in the following to the detailed comments for which we are in agreement and of which the most part come from a confusion which will be corrected by clarifications in

the article.

Thank you for your comments,

Authors

Answers to the detailed comments:

1-5 will be updated in the preprint regarding the comments.

8. We agree on the GOE trigger. The confusion can be corrected in the preprint and the reference added.

9. The comment about nitrogen is relevant. Its chemistry is not constrained but not negligible, this will be corrected.

11. It is indeed undeniable that the surface temperature will impact the biological activity at the surface. Nevertheless the approach here does not contradict these facts. It tries to put forward the effect only on the atmosphere which has not been put forward until now, whereas the effect on the biological activity is better known. This methodological confusion in the article can also be clarified following this comment.

[Figure]

Clim. Past Discuss., author comment AC2
https://doi.org/10.5194/cp-2021-150-AC2, 2022

[Figure]

**Reply on RC2**

Adam Yassin Jaziri et al.
* * *
Author comment on "Dynamics of the Great Oxidation Event from a 3D photochemical-climate model" by Adam Yassin Jaziri et al., Clim. Past Discuss., https://doi.org/10.5194/cp-2021-150-AC2, 2022
* * *
We thank the referee for constructive and useful comments. We have modified the manuscript taking all comments into account.

**Photochemical modeling**

*"Summary comments: I'm really happy to see some 3-D work on the oxygen photochemistry, but I think the presentation of the 3-D results is quite lacking, so the (no-doubt) interesting results from the modelling work are not made available to the reader! Also, much of the work presented and used in the manuscript is actually from a 1-D model, and clearer distinction of this needs to be made. Statements about code availability, sufficient explanation of the methods to realistically enable reproduction, and an archive of model output are needed.*

*There is not sufficient information in the methods to allow the work to be repeated. Via online repositories of code and/or machine-readable supplementary information, it is now possible to make modelling work genuinely reproducible. This should be done. If for some reason this cannot be done, an exceptional justification is necessary. There is no statement about the availability of the codes used in the manuscript, which is necessary (are the codes open source? If not, how can they be acquired?). Complete sets of inputs need to be provided (what are the actual boundary conditions for each model run – e.g. what fluxes of which reductants are used?). A machine readable version of Appendix A should be included. Some details are vague (e.g. line 108 speaks of an eddy diffusion coefficient, whereas typically a vertical profile of these are used) – all details need to be clear.*

*There no archive of model outputs, which should be important, especially for the GCM runs which are very costly to produce. I would recommend, in the strongest terms, that authors make their data (model output) available following FAIR principles (https://www.go-fair.org/fair-principles/). Providing this to other researchers will add significant value to the manuscript."*

We agree that additional plots showing horizontal/time evolution of different variables from 3D simulations would be useful to highlight consequences of the atmospheric dynamics and change of insolation. We have added ozone plots (the most relevant species for this study) variations from 3D simulations. In addition, we will provide a link to the data results and to the code, which is already open source. It will be done as inTurbet et al. 2021, and included according to further editor's instructions. The distinction between 1D and 3D models are specified in the paper each time we refer to a model, and also in the caption of the figures.

*"The photochemical models use fixed mixing ratio boundary conditions at the surface. An excellent recent paper by Gregory et al. (2021) compares fixed mixing ratio and fixed flux boundary conditions, and concludes that fixed flux boundary conditions are preferable. This should be discussed, and the approach used here justified in that context, including discussion of any bias that fixed mixing ratio boundary conditions might introduce."*

We fixed the O2 mixing ratio at the first atmospheric level and we explored its impact on the atmospheric equilibrium states for a range of mixing ratio from $10^{-7}$ to $10^{-3}$ while Gregory et al. 2021 fixed the O2 surface flux. Reference to this work and explanation of our choice is added in the model description. Indeed, the two approach are different, and some additional tests with our 3D model seem to show different results. This cannot only be explain only by Gregory et al. 2021 since the 3D effects due to the fixed boundary condition at the surface could be significant depending on the two approach. This effect could be deeply study in another paper focus on this analysis which is very interesting but need its own study.

*"You note that the ozone column is large in 1-D (line 150). Is this a genuine difference between 1-D and 3-D approaches, or is your zenith angle simply poorly tuned in the 1-D model, given knowledge of the 3-D results?"*

The zenith solar angle for 1D calculations has been set up to 60 degrees, a quite usual approximation. There is no perfect value to use in 1D and it is not simple to tune it. So it would be arbitrary to use another zenith angle, which is something which could be studied deeply apart. But, complementary simulations at a zenith solar angle of 40 degrees shows only tiny differences for ozone compare to the 3D. This has been added in the manuscrit.

*"The justification for using methane fluxes as a proxy for oxygen fluxes (line 163-4) is not clearly justified. Use of molecular oxygen to oxidize CO (which I suspect you include as a boundary condition) would plainly be an oxygen loss. You need to be clearer about why this is excluded."*

We focus on methane fluxes as a proxy for oxygen fluxes because oxygen loss by oxidizing CO is recycled while it is not with the methane oxidation. There is no sources/losses of CO at the surface, so the CO oxidation 2CO+O2-> 2CO2 and the reverse reaction have a null balance for O2, contrary to methane oxidation. This makes the global loss described by methane loss dominated by the methane oxidation. We developed this explanation.

*"Overall, I found the presentation of the results from the 3-D modelling a disappointment. There is only one figure which shows anything other than global means (Fig 5, which shows vertically resolved zonal means). Surely there are actually some interesting results in 3-D?! Conversely, if there really are not, then it would be a great service to the community to robustly make the case that 3-D models are quite unnecessary for this problem, so that no-one else need bother!*

*To be clear here: I strongly recommend more detailed presentation of the results from the GCM. For example, the questions that I would start with are: What is the model climatology of ozone and water vapour above the tropopause, and how does this compare to climatology? What is the model Brewer-Dobson circulation, and how does this compare to climatology? Note that this may well affect the ozone distribution, and how the rates differ from 1-D. How do these change with model temperature? Is there any feedback between changed ozone levels and the climate?"*

Our analysis in this paper is essentially limited to the calculation of the oxygen loss to refine the oxygen evolution during the GOE. We added ozone zonal mean maps and compared to the current Earth with and without continent in order to highlight the climatology related to ozone at this period. We discuss the seasonal evolution of the ozone column density.

*"I am a little surprised by how cold the stratosphere gets. Does your radiation scheme include near-IR absorption by methane, which has been shown to cause significant warming in 1-D models (Byrne & Goldblatt, 2015)? Also, note that at line 205 you refer to 220 and 280K, whereas Figure 12 shows results for 260 and 280K."*

We used HITRAN 2012 lines for the CH4 near-IR absorption, as Byrne & Goldblatt 2015.

*"In appendix B, dominant photochemical pathways are presented. However, there is no discussion of the methods by which they were found, which is necessary. As with all the methods, this discussion should be sufficient for these to be reproducible, including reference to the codes used. There is no substantial discussion of these in the text of the manuscript, which would be important to contextualize them."*

Due to the complexity of the chemistry, the pathway derived in appendix B are empirical and it should be more hightlited, which as been corrected now in the manuscript.

*"There has been prior work that has used 3-D models to examine atmospheric oxygen photochemistry, which should be referenced (e.g. Cooke, 2022)."*

The recent paper of Cooke et al. 2022 should be indeed referenced in the manuscrit. It was published after our submission and we planned to include it in the revised manuscript.

Minor comments have been mostly corrected

**Box model**

*"Re original model by Goldblatt et al (2006). Unfortunately, the polynomial fit constants were truncated in the manuscript, but more decimal places are needed. Use p = { 0.0030084, -0.1655405, 3.2305351, -25.8343054, 71.5397861} to reproduce the result shown (I'm sorry about this!). Note also that the value of hydrogen escape coefficient was calculated incorrectly in the original manuscript; the correct value is s = 3.7e-5, but that doesn't have a very big effect (see Goldblatt, 2008, if a reference is needed for that; there is also a bonus chapter on the model there which might be interesting to you)."*

We tried with the updated "p" value, which did not change the results and the conclusion.

*"The original model is modified with parameters to make the methane oxidation rate smaller and the primary productivity bigger. I understand the motivation of the former, but not the latter, given that N is a variable in the original model, not a constant or function. Why not simply talk about increasing N? Results for changing N are presented in the original manuscript, so the result that higher productivity gives more atmospheric oxygen is not novel."*

As referred at the beginning of part 4, indeed, a change in N with higher productivity giving more atmospheric oxygen is not novel. We have chosen in this manuscript to adapt the equation to have a temporal evolution on the parameter N. It combines previous studies to show analytically the overshoot that could be then compared to the new approach with the atmospheric losses variability.

*"Please be clear about what results came from your 3-D GCM, and which came from your 1-D column model (a 1-D model is not a GCM: do not refer to it as such! See lines 272 & 284 and Fig 15). Note also, line 272, 1-D and 3-D results were shown to be similar, not "identical"."*

We adjusted the notation to specify each time if the model is 1D or 3D in order to be clearer.

*"Please explain why oxygen levels are shown to be constant at low oxygen and high r in Figs 15 & 16. My guess is that this is bogus: that you have used a constant value of your oxidation rate below 10-7 O2, where you do not have model results, which would be erroneous. Either some extrapolation of results is needed, or do not show your results at such low oxygen levels."*

The constant level of oxygen at low oxygen and high r (same as high methane) in Figs 15 & 16, is related to the apparent asymptotic convergence of the atmospheric losses at those levels of oxygen and methane (Fig 14). Below 1e-7, the values follow a log-log linear asymptotic trend with a slope calculated with the atmospheric loss at 1e-3 CH4 vmr, level which is not reached. This asymptote is not well explained but is also seen in Zahnle et al. 2006. This is now more detailed in the manuscript.

*""We model a long variation over 400 million years of the primary productivity, parameter $\alpha_N$, with a first phase of over-production before a return to the initial production. We represent Figure \ref{dynt} the reference evolution for constant $\alpha_N$ equal to*

*1, as well as two different intensities of over-productivity with a maximum at 2 and 10 times the initial productivity. The reference model is in good agreement with the results of \cite{Goldblatt} and \cite{Claire}. We observe a variable over-oxygenation depending on the intensity of the primary productivity variation, but also an over-abundance of methane. The primary productivity corresponds to a photosynthetic production of oxygen but also of organic matter, transformed then into methane by the methanogens. Consequently, the production of methane is increased as well as that of oxygen. Such an over-abundance of methane is not highlighted by previous studies, nor by the geological record. It is difficult to constrain the amount of methane at that time. This scenario is therefore possible, although one could also imagine that oxygen enrichment of the atmosphere limited the conversion of organic matter into methane. A conversion that is carried out by methanogens, developing more favorably in a reducing environment. This negative feedback is not taken into account in the biosphere model."*

*You state (line 299) that a reduced methane flux at higher oxygen levels is not treated in the model. This is functionally incorrect, as the methane flux is a very strong function of oxygen via parameters d and g (while the words used in the paper were about organic carbon available to methanogens, and consumption by methanotrophs, the effect is functionally as you seek!)."*

Indeed, we removed this wrong comment.

*"The realism of the scenario of altering productivity and methane oxidation rate, and the relative timing of these (e.g. Fig 20) need to be justified. What motivates the very long period of high productivity, starting before the glaciations? Why do you show four glacial periods (see, for example, Gumsley, 2017)? In the glacial periods, you keep productivity the same, whereas most people assume that synglacial productivity is much reduced. This is important, because reduced productivity would more than offset the effect of slower methane oxidation."*

The scenario Fig 20 is not fundamentally realistic but puts forward different effects that could have played a role on the oxygen and methane evolution during the GOE. It has been clarified in the manuscript.

*"You assert that methane would produce a substantive greenhouse effect, and reduction of this could lead to glaciation (e.g. lines 33, 324, 341), but these comments are not supported. There has been some historical misunderstanding about the strength of this, owing to erroneous results twenty years ago. Recent papers should be consulted, based on which a quantitative estimate can be made of how strong an impact this is (e.g. Haqq Misra et al, 2008; Byrne and Goldblatt, 2014; Byrne and Goldblatt, 2015). I am rather doubtful that the radiative forcing from methane would exceed that from changes in carbon dioxide (not modelled!). I am extremely doubtful that methane would have cause cycles, and have a stronger effect than known cycles or carbon dioxide (e.g. Mills et al, 2011) – though, of course, if you could make the case, then I would be fascinated!"*

Indeed, such diminution of methane could significantly affect the surface temperature. As it is shown in Charnay et al. 2020, a diminution of a factor 10 in methane decreases the mean surface temperature of approximately 4K. Furthermore, Sauterey et al. 2020 show that the diminution of CH4 combined with CO2 carbonate-silicate cycle regulation favors the triggering of an ice age. These details have been added in the manuscript.

Minor comments have been mostly corrected

The authors

**References:**

Turbet, M., Bolmont, E., Chaverot, G. et al. Day–night cloud asymmetry prevents early oceans on Venus but not on Earth. Nature 598, 276–280 (2021). https://doi.org/10.1038/s41586-021-03873-w

Zahnle, K., Claire, M., & Catling, D. (2006). The loss of mass☐independent fractionation in sulfur due to a Palaeoproterozoic collapse of atmospheric methane. Geobiology, 4(4), 271-283.

Charnay, B., Wolf, E. T., Marty, B., & Forget, F. (2020). Is the faint young Sun problem for Earth solved?. Space Science Reviews, 216(5), 1-29.

Sauterey, B., Charnay, B., Affholder, A., Mazevet, S., & Ferrière, R. (2020). Co-evolution of primitive methane-cycling ecosystems and early Earth's atmosphere and climate. Nature communications, 11(1), 1-12.